# Wave Climate along Calabrian Coasts

**Giandomenico Foti** [1,*] , **Giuseppe Barbaro** [1] , **Giovanni Besio** [2] , **Giuseppina Chiara Barillà** [1] , **Pierluigi Mancuso** [3] and **Pierfabrizio Puntorieri** [1]

1    Department of Civil, Energy, Environmental and Material Engineering (DICEAM), Mediterranea University of Reggio Calabria, Via Graziella loc. Feo di Vito, 89122 Reggio Calabria, Italy; giuseppe.barbaro@unirc.it (G.B.); chiara.barilla@unirc.it (G.C.B.); pierfabrizio.puntorieri@unirc.it (P.P.)
2    DICCA Department, Genoa University, Via Montallegro 1, 16145 Genoa, Italy; giovanni.besio@unige.it
3    Public Works Department, Calabria Region, Viale Europa loc. Germaneto, 88100 Catanzaro, Italy; pierluigi.mancuso@regione.calabria.it
*    Correspondence: giandomenico.foti@unirc.it

**Abstract:** The wave climate is highly variable temporally and spatially, depending mainly on the atmospheric conditions and on fetch extensions. Wave climate is one of the main causes of coastal erosion processes, together with anthropogenic pressure and with coastal and river sedimentary balance. Therefore, a detailed spatial and temporal knowledge of wave climate is very important in managing coastal areas and in planning coastal defense works. This paper describes an analysis of the wave climate carried out along the Calabrian coasts in over 50 areas, each of them covering an average of 15 km of coastline. For each area, over 40 years of wave data were analyzed to calculate over 20 parameters, representative of annual and seasonal average and maximum wave conditions. The large number of areas is related to the geomorphological and climatic complexity of Calabria. This analysis mainly highlighted that the two Ionian and Tyrrhenian coasts are very different from the wave climate point of view. Indeed, the Ionian coast is heavier in ordinary wave conditions, while the Tyrrhenian coast is heavier in extreme wave conditions.

**Keywords:** wave climate; wave data; wave parameter; maximum significant wave height; energy flux; frequent conditions; extreme conditions





## 1. Introduction

One of the main mechanisms of wave motion generation is the transfer of energy from the atmosphere to the sea surface through the wind. Therefore, the wave motion generated by the wind is closely related to atmospheric conditions and is particularly variable in space and time [1–7]. Another important factor causing spatial variations in the wave climate is the fetch length. Therefore, the wave climate generally depends on the wind speed and duration, on the airflow stability, on the bathymetry, and on the length of the fetch where the wind acts [8–12]. The dependence between wave climate and fetch length also implies a dependence between wave climate and coastal morphology [13–16]. This dependence is particularly important both in closed basins, such as the Mediterranean Sea, where the fetches vary considerably as the direction changes, and for irregular coastal morphologies, with gulfs, straits, and promontories.

To analyze the wave climate, it is necessary to have time series of wave data long enough to represent a statistically significant sample. The main types of wave data are those measured in situ by buoys or those reconstructed by numerical modeling [17–21]. Buoy data have the advantage of being real data but also have the disadvantage of being spatially limited. For example, in Italy, there are only about fifteen buoys for over 8000 km of coastline, and in Calabria, a region of southern Italy, there are only two buoys for over 700 km of coastline. Furthermore, these buoys have time series varying from a few years up to 25 years, with a recent lack of records for seven years. On the other hand, hindcast

model data have the disadvantage of not being real data but have the advantage of greater spatial and temporal coverage.

Wave climate is determined by many parameters, the main ones being significant wave height, peak and mean period, mean wave direction, mean energy flux, main sector and, where present, secondary and tertiary sectors [22]. Wave climate, through the action of sea storms, is the main cause of coastal flooding [23,24], together with intense rainfall events [25,26], especially when these events are concurrent [27–29]. Additionally, wave climate is one of the main causes of coastal erosion [30–32]. Other important causes are the alteration of coastal sedimentary balance, which is mainly related to longshore and river transport [33–39], and anthropogenic pressure in coastal and river areas [40–46], through the destruction of dune systems [47–50], the withdrawal of river sediment [51] and the construction of ports and coastal and river structures [52–57]. Therefore, the correct and detailed knowledge of wave climate is very important to manage coastal areas and to plan coastal defense works [58].

The paper describes an analysis of the wave climate carried out along the Calabrian coasts starting from the wave data from the last 40 years, available in the database developed by the MeteOcean group of the University of Genoa. The analysis was carried out in more than 50 points of the MeteOcean database grid and the values of over 20 wave parameters were calculated to analyze the spatial and temporal variations of the wave climate in a very complex case such as the Calabrian one. Indeed, Calabria is a region in southern Italy that represents an interesting case study due to its notable coastal length and due to the considerable variability of climatic and geomorphological conditions.

## 2. Materials and Methods

### 2.1. Site Description

Calabria is an interesting case study of wave climate analysis due to its geographical position and due to some geomorphological and climatic peculiarities.

From the geographical position point of view, Calabria is a region of southern Italy that is in the center of the Mediterranean Sea and is a peninsula within the Italian peninsula, with a total coastal length of about 750 km (Figure 1). In fact, it is bathed by the sea to the south, east, and west, and is connected to the Italian peninsula only to the north, through a strip of land about 75 km long. The eastern coast of Calabria is located on the Ionian Sea; this sea is bordered by Greece to the east and by the Libyan Sea to the south and is characterized by depths of over 5000 m and by fetches of the order of hundreds of kilometers, up to over 1000 km. The western coast of Calabria is located on the Tyrrhenian Sea; this sea is bordered by Corsica and Sardinia to the west, by Sicily to the south, and by the Italian peninsula to the north and is characterized by depths of about 3800 m and by fetch lengths of order of hundreds of kilometers, up to over 700 km. Within the Ionian Sea there are two gulfs, Taranto and Squillace. The Gulf of Taranto is in the northern part of Calabria between the northern border of Calabria and Punta Alice to the south and is characterized by depths of about 1500 m and by fetch lengths up to about 130 km. The Gulf of Squillace is in the central part of Calabria between Capo Rizzuto and Punta Soverato and is characterized by depths over 500 m and by fetch lengths up to about 60 km. On the other hand, within the Tyrrhenian Sea, there are three gulfs, Policastro, Sant'Eufemia, and Gioia Tauro. The Gulf of Policastro is in the northern part of Calabria and is characterized by depths over 500 m and by fetch lengths up to about 30 km. The Gulf of Sant'Eufemia is in the central part of Calabria and is characterized by depths of about 200 m and by fetch lengths up to about 30 km. The Gulf of Gioia Tauro is in the southern part of Calabria and is characterized by depths over 300 m and by fetch lengths up to about 40 km. Finally, in its southern part, Calabria is separated from Sicily by the Strait of Messina. This Strait is characterized by depths of up to 2000 m and by fetches of varying lengths between 3 and 30 km.

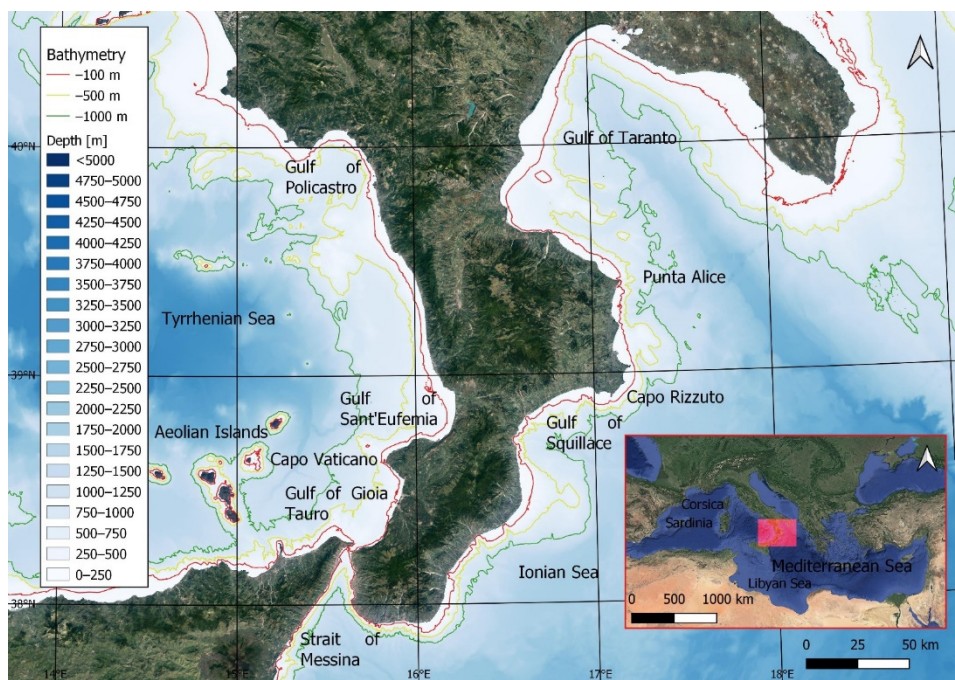

**Figure 1.** Calabria region (large panel). Geographical position of Calabria region in the center of Mediterranean Sea (small panel).

From the bathymetry point of view, there are considerable variations between the two coasts. Indeed, the analysis of the bathymetry available on the European Marine Observation and Data Network (EMODnet) portal (https://www.emodnet-bathymetry. eu/, accessed on 5 April 2022) showed that the Tyrrhenian Sea has generally shallower bottoms and less slope than the Ionian Sea offshore. On the other hand, nearshore, the Ionian Sea has less slope than the Tyrrhenian Sea. In detail, in the center of the Gulf of Taranto, there is a submarine canyon that exceeds 1500 m in depth and the bathymetric −100 m is located at distances between 5 and 15 km from the coast, except in front of point I5 where this distance is about 500 m. Additionally, in the coast between Punta Alice and point I29, the bathymetric −100 m is almost always located about 5 km from the coast, except in front of points I18, I23, I24 and I26 where this distance is a few hundred meters. In these last three points, there are submarine canyons in front of the mouths of some rivers. In the Strait of Messina and in the adjacent coasts, between points I28 and T2, the seabed is steeply sloping. In fact, the bathymetric −100 m is almost always located a few hundred meters from the coast, while the bathymetric −1000 m in the steeper part is located less than 3 km from the coast. In the Tyrrhenian Sea, the bathymetric −100 m is located less than 4 km from the coast except for the Gulf of Sant'Eufemia and in front of points T15 and T16. Finally, about 80 km in front of the Gulf of Gioia Tauro to the west are the Aeolian Islands, a small archipelago of volcanic islands that partially cover the waves coming from the west direction. From the geomorphological point of view, 90% of the Calabrian territory is mountainous or hilly. The main massifs are Pollino, Sila and Aspromonte, all with a maximum altitude of the order of 2000 m, and the Catena Costiera, which is located a short distance from the northern Tyrrhenian coast and has a maximum height of over 1500 m. In addition, there are some promontories and the main are Capo Rizzuto, on the Ionian coast, and Capo Vaticano, on the Tyrrhenian coast. The main coastal plains are that of Sibari, on the Ionian coast in the Gulf of Taranto, and those of Lamezia Terme in the Gulf of Sant'Eufemia and of Gioia Tauro in the homonymous gulf, both on the Tyrrhenian coast. Regarding the coastline, there is an alternation between sandy and pebbly beaches and high coasts.

From the climatic point of view, the Calabrian climate is strongly influenced by the geomorphological variability described above. The mountainous areas have a typical

mountain climate, with frequent snow during the winter. Instead, the coastal areas have a Mediterranean climate with significant differences in rainfall and temperatures between the two coasts. In fact, the Tyrrhenian coast is cooler and rainier than the Ionian one. The greatest rainfall occurs mainly in winter and autumn and is significantly reduced in the summer months, varying between 1400 and 1800 mm per year in the mountainous areas, between 700 and 1000 mm per year on the Tyrrhenian coast and around 500 mm per year on the Ionian coast. The sea water temperature reaches the highest value in July and August with 26 °C, it remains around 22–23 °C until October and then decreases to 14 °C in winter. The high sea water temperature in the autumn months favors the formation of particularly intense atmospheric disturbances, which sometimes become a kind of hurricane, also called medicane (Mediterranean hurricane) or tropical-like cyclones (TLC), as happened in 2015 in Bruzzano, on the southern Ionian coast [59]. Finally, the Ionian coasts are mainly exposed to the winds of Scirocco, south-east, and Grecale, north-east, while the Tyrrhenian coasts are mainly exposed to the winds of the Mistral, north-west. These differences, together with the different fetches' lengths between the two seas and the various gulfs, lead to a remarkable variability of sea conditions between the two coastal areas.

*2.2. Methodology*

The wave climate along Calabrian coasts was analyzed starting from the wave data from the last 40 years, available in the database developed by the MeteOcean group of the University of Genoa (http://www3.dicca.unige.it/meteocean/hindcast.html, accessed on 15 February 2022). This group has performed a re-analysis of atmospheric and wave conditions, producing an hindcast database that starts from January 1979 until today. This database has been reconstructed from the Climate Forecast System Reanalysis (CFSR) database through a numerical model that consists of a meteorological model for the reanalysis and the simulation of winds and atmospheric fields and in a third-generation model for the description of the generation and the propagation of wind and swell waves in the Mediterranean basin. In detail, the wind forcing has been provided by the 10 m wind fields obtained using the non-hydrostatic mesoscale model Weather Research and Forecasting with the Advanced Research solver (WRF-ARW). The wave conditions were analyzed using the third-generation wave model WavewatchIII. Additionally, the Mediterranean basin has been discretized into a regular grid with a resolution of $0.1273 \times 0.09$ degrees, corresponding almost to 10 km at the latitude of 45° N [60–65]. To consider the significant geographical differences between the Calabrian coasts described above, some points of the grid have been chosen that characterize the wave climate along the various coastal areas. In detail, over 50 points were chosen, so the distance along the coast between one point and another is about 15 km (Figure 2). Each point was identified with the suffix I for the points of the Ionian Sea and with the suffix T for the points of the Tyrrhenian Sea, both followed by an increasing number in a clockwise direction, and was characterized by the time series of the order of hundreds of thousands of sea states, from a minimum of 330,000 to a maximum of 350,000, starting on 1 January 1979 and ending on 31 December 2021, with hourly data. All points are located on deep water, as no points are located at depths less than 100 m, except for point I20, which is located at depths of 87 m. Additionally, for each sea state, significant wave height, mean and peak periods and wave direction are available.

For each point, the entire time series were grouped into sectors of 10° each and the following parameters were calculated:

1. Maximum significant wave height $h_{s,max}$;
2. Average significant wave height $h_s$, average peak period $t_p$ and average mean period $t_m$;
3. Frequency $f$ of each sector;
4. Average annual energy flux $\Phi$ of each sector;
5. Average annual energy flux $\Phi_t$;
6. Main, secondary and tertiary sectors;

7. Significant wave height of fixed return period ($h_{s1}$, corresponding to return period of 1 year, and $h_{s100}$, corresponding to return period of 100 years, to consider both frequent and rare events) and their difference $\Delta h_{s1-100}$;
8. Characteristic parameters $u$, $w$, $a_{10}$, $b_{10}$.

In addition, a seasonal analysis was performed where the following parameters were calculated (where the i-th season has been indicated with subscripts $W$ for winter, $SP$ for spring, $SU$ for summer and $A$ for autumn):

9. Seasonal maximum significant wave height $h_{s,max,i}$;
10. Seasonal average significant wave height $h_{s,i}$;
11. Seasonal average annual energy flux $\Phi_{t,i}$;
12. Seasonal main sector $MS_i$.

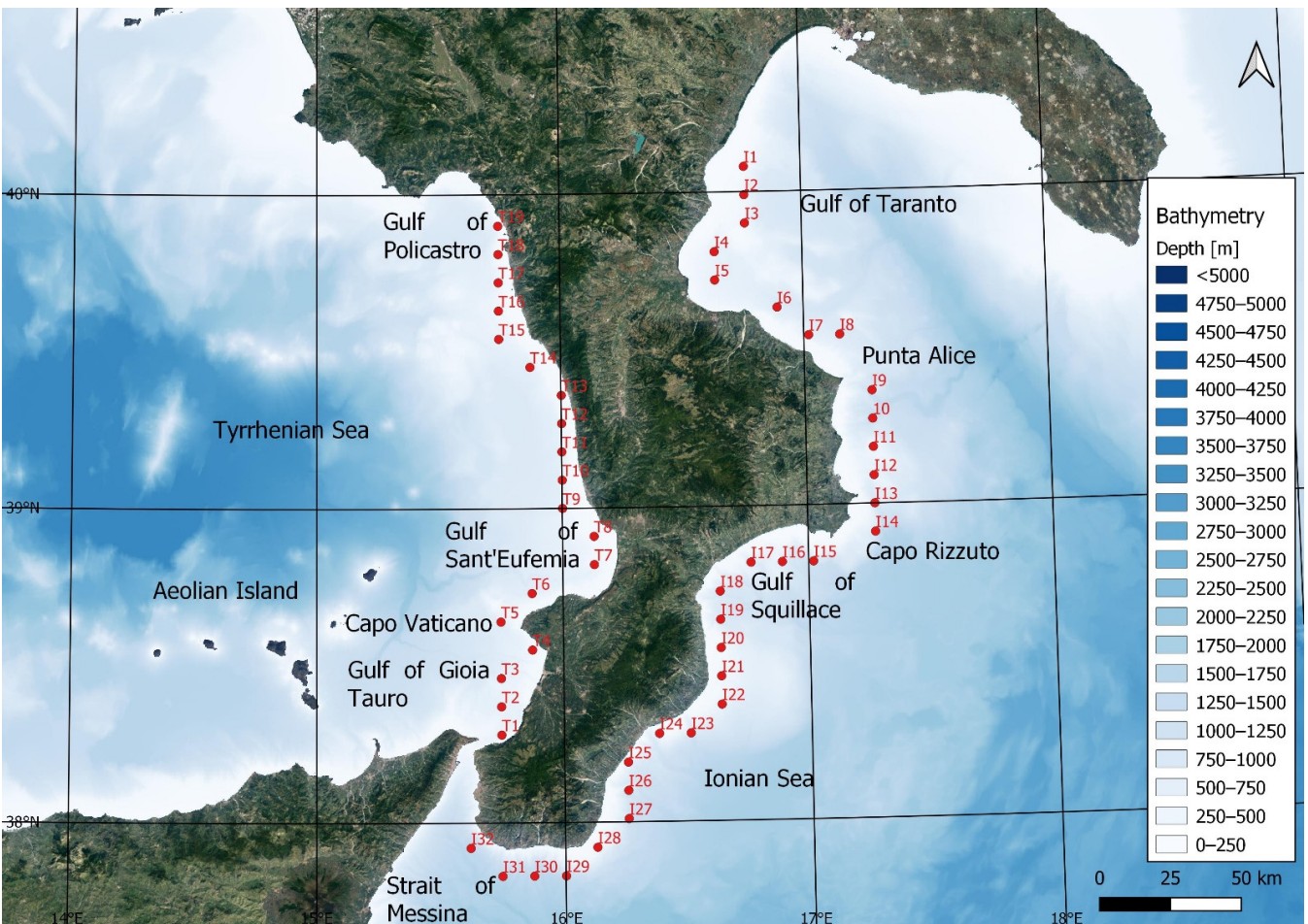

**Figure 2.** Calabrian sample areas, identified with the suffix I for the points of the Ionian Sea and with the suffix T for the points of the Tyrrhenian Sea, both followed by an increasing number in a clockwise direction.

The frequency of each sector is the ratio between the number of sea states from that sector and the total number of registered sea states. The average annual energy flux of each sector is the sum of the average annual energy flux of each sea state from that sector, which depends on the specific gravity of the water, on the peak period, on the square of the significant wave height and on the frequency. So, the average annual energy flux of each point is the sum of the average annual energy flux of each sector. The main sector is the one with the greatest energy flux value, while the secondary and tertiary sectors are those with the second and third highest energy flux values. The criterion adopted to define the secondary and tertiary sectors is as follows: sectors characterized by an

increasing mean energy flux compared to the adjacent sector and not contiguous to the main sectors (in the case of secondary sector) and not contiguous to the secondary sector (in the case of the tertiary sector). So, the secondary and tertiary sectors are not always present. The significant wave height of fixed return period was estimated reversing the following formula [63]:

$$T = (b_{10} \, (1.12 - 0.12 \, h_s/a_{10})) \times (exp \, (h_s/w)^u)/(1 + u \, (h_s/w)^u) \tag{1}$$

where the parameters $u$ and $w$ describe a Weibull-type distribution of the probability of exceeding a fixed significant height threshold, while the parameters $a_{10}$ and $b_{10}$ are related to the equivalent triangular storm model (ETS) [66]. Among the various distributions of the probability of exceeding, Weibull was chosen as appropriate for the Mediterranean Sea [66,67], and applied in Italy, for example, by [68,69]. The reliability of the Weibull distribution was tested by calculating the coefficient of determination R-squared [70]. ETS is a model that associates an equivalent triangular storm to each real sea storms. The height $a$ of the equivalent triangular storm is equal to the maximum significant wave height of the real sea storm, and the base $b$ of the equivalent triangular storm is obtained imposing equality between the maximum expected wave height during the equivalent triangular storm and during the real sea storm. Therefore, $a_{10}$ and $b_{10}$ represent, respectively, the average values of $a$ and $b$, referring to a sample of the $N$ stronger real sea storms, with $N$ equal to 10 for the number of years of available data. Furthermore, the parameters $u$, $w$, $a_{10}$ and $b_{10}$ are characteristic of the analyzed location.

## 3. Results

Tables 1–4 show a summary of the calculated parameters. In detail, Table 1 shows the coastal municipalities present in the coastal area corresponding to each point and the values of $h_{s,max}$, $h_s$, $t_p$, $t_m$, $h_{s1}$, $h_{s100}$ and $\Delta h_{s1-100}$ calculated for each point. Table 2 shows the values of average annual energy flux $\Phi_t$, the main $MS$, secondary $SS$ and tertiary $TS$ sectors and the corresponding values of energy flux $\Phi_{MS}$, $\Phi_{SS}$ and $\Phi_{TS}$, respectively, the ratio between the main sector energy flux value and the total energy flux value $\Phi_{MS}/\Phi_t$ and the characteristic parameters $u$, $w$, $a_{10}$ and $b_{10}$. Table 3 shows the values of seasonal maximum significant wave height $h_{s,max,i}$ and of seasonal average significant wave height $h_{s,i}$. Table 4 shows the values of seasonal average annual energy flux $\Phi_{t,i}$, and of seasonal main sector $MS_i$. The i-th season has been indicated with subscripts $W$ for winter, $SP$ for spring, $SU$ for summer and $A$ for autumn. The maximum values of $h_{s,max}$ are between 4.91 m at point I5, located in the Gulf of Taranto, and 9.82 m at point T9, located in the central-northern Tyrrhenian Sea. The average value of $h_{s,max}$ is 7.33 m, and this parameter is characterized by a considerable spatial variability between the two macro areas Ionic and Tyrrhenian and within each macro area. Indeed, in the Ionian coast, the $h_{s,max}$ values are always lower than 7.5 m, while in the Tyrrhenian coast, the $h_{s,max}$ values are always higher than 8, with the exception of three points (T1, T4, T19). Spatial variability is then analyzed in detail by proceeding clockwise starting from the Ionian coast. The lowest $h_{s,max}$ values are observed in the innermost part of the Gulf of Taranto, in the points between I1 and I5, with values always lower than 6 m. From point I5 towards the east, the $h_{s,max}$ values increase up to about 7 m at point I8, located near Punta Alice at the mouth of the Gulf of Taranto. Below, between Punta Alice and Capo Rizzuto in the points between I9 and I14, the highest $h_{s,max}$ values of the whole Ionic macro area are observed, with values between 7 and 7.5 m. Following, within the Gulf of Squillace, the $h_{s,max}$ values gradually decrease to just over 6 m at point I18 near Punta Soverato. From this point southwards to point I22, the $h_{s,max}$ values increase over 7 m. Then, there are three points, from I23 to I25, with very similar $h_{s,max}$ values of about 6.4 m, followed by a new growth up to point I27, with $h_{s,max}$ values up to almost 7.4 m. From this point towards the west, up to the southern mouth of the Strait of Messina, the $h_{s,max}$ values decrease to just over 6.3 m. At point T1, near the northern mouth of the Strait of Messina, the $h_{s,max}$ value is about 6.7 m and is the lowest of the entire Tyrrhenian coast. Inside the Gulf of Gioia Tauro, between points T2

and T4, the $h_{s,max}$ values are between 7.8 and 8.3 m and decrease towards the north. On the promontory of Capo Vaticano, the $h_{s,max}$ values exceed 9 m and then decrease within the Gulf of Sant'Eufemia up to just over 8.5 m. At the point immediately north of this is observed the maximum $h_{s,max}$ value of the whole of Calabria, over 9.8 m. Subsequently, the $h_{s,max}$ values decrease towards the north remaining above 9 m and then decrease to 8.3 m at point T13. The three points following it, from T14 to T16, have very similar $h_{s,max}$ values of about 9.5 m. In the next two points, T17 and T18, the $h_{s,max}$ values are lower than the previous ones but are still high, between 8.3 and 8.9 m. Finally, at point T19 the $h_{s,max}$ value is 7.15 m, much lower than the other Tyrrhenian values but this point is inside the Gulf of Policastro.

**Table 1.** Coastal municipalities present in the coastal area corresponding to each point and the values of maximum significant wave height $h_{s,max}$, average significant wave height $h_s$, average peak period $t_p$, average mean period $t_m$, significant wave height of return period of 1 year $h_{s1}$, significant wave height of return period of 100 years $h_{s100}$, and difference between significant wave height of return period of 1 and 100 years $\Delta h_{s1-100}$, calculated for each point. Legend: the higher values are in red, and the lower values are in green.

| Point | Coastal Municipality | $h_{s,max}$ [m] | $h_s$ [m] | $t_p$ [s] | $t_m$ [s] | $h_{s1}$ [m] | $h_{s100}$ [m] | $\Delta h_{s1-100}$ [m] |
|---|---|---|---|---|---|---|---|---|
| I1 | Rocca Imperiale, Montegiordano | 5.68 | 0.56 | 4.57 | 3.87 | 3.67 | 6.68 | 3.01 |
| I2 | Roseto Capo Spulico | 5.72 | 0.58 | 4.51 | 3.82 | 3.73 | 6.62 | 2.89 |
| I3 | Amendolara, Albidona, Trebisacce | 5.88 | 0.61 | 4.43 | 3.76 | 3.69 | 6.23 | 2.54 |
| I4 | Villapiana, Cassano allo Ionio | 5.57 | 0.50 | 4.22 | 3.57 | 3.19 | 5.44 | 2.25 |
| I5 | Corigliano-Rossano to the Trionto River mouth | 4.91 | 0.43 | 3.94 | 3.36 | 2.76 | 4.71 | 1.95 |
| I6 | Corigliano-Rossano from the Trionto River mouth to the east, Crosia, Calopezzati, Pietrapaola, Mandatoriccio | 6.04 | 0.54 | 4.30 | 3.62 | 3.46 | 5.88 | 2.42 |
| I7 | Scala Coeli, Cariati | 6.18 | 0.57 | 4.37 | 3.67 | 3.57 | 6.03 | 2.46 |
| I8 | Crucoli, Cirò, Cirò Marina up to Punta Alice | 6.87 | 0.71 | 4.66 | 3.94 | 4.11 | 6.86 | 2.75 |
| I9 | Cirò Marina from Punta Alice to the south | 7.42 | 0.79 | 4.94 | 4.19 | 4.50 | 7.58 | 3.08 |
| I10 | Melissa, Strongoli, Crotone to the Neto River mouth | 7.44 | 0.79 | 4.99 | 4.22 | 4.49 | 7.54 | 3.05 |
| I11 | Crotone from the Neto River mouth to the port | 7.34 | 0.80 | 5.04 | 4.27 | 4.53 | 7.63 | 3.10 |
| I12 | Crotone from the port up to Capo Colonna | 7.23 | 0.81 | 5.11 | 4.33 | 4.55 | 7.67 | 3.12 |
| I13 | Crotone from Capo Colonna to the south, Isola Capo Rizzuto up to Capo Cimiti | 7.02 | 0.84 | 5.18 | 4.39 | 4.58 | 7.64 | 3.06 |
| I14 | Isola Capo Rizzuto from Capo Cimiti up to Capo Rizzuto | 7.04 | 0.92 | 5.22 | 4.43 | 4.77 | 7.82 | 3.05 |
| I15 | Isola Capo Rizzuto from Capo Rizzuto up to Le Castella | 6.69 | 0.82 | 5.08 | 4.31 | 4.42 | 7.46 | 3.04 |
| I16 | Isola Capo Rizzuto from Le Castella to the west, Cutro, Belcastro, Botricello, Cropani, Sellia Marina to the Uria River mouth | 6.45 | 0.76 | 4.99 | 4.23 | 4.16 | 7.06 | 2.90 |
| I17 | Sellia Marina from Uria River mouth to the south-west, Simeri Crichi, Catanzaro, Borgia, Squillace, Stalettì up to Punta di Stalettì | 6.28 | 0.68 | 4.92 | 4.17 | 3.92 | 6.93 | 3.01 |
| I18 | Stalettì from Punta di Stalettì to the south, Montauro, Montepaone, Soverato up to Punta Soverato | 6.11 | 0.67 | 4.90 | 4.18 | 3.90 | 7.11 | 3.21 |
| I19 | Soverato from Punta Soverato to the south, Satriano, Davoli, San Sostene, Sant'Andrea Apostolo dello Ionio | 6.11 | 0.69 | 5.00 | 4.24 | 3.96 | 7.06 | 3.10 |
| I20 | Isca sullo Ionio, Badolato, Santa Caterina dello Ionio | 6.51 | 0.71 | 5.11 | 4.32 | 4.19 | 7.46 | 3.27 |
| I21 | Guardavalle, Monasterace | 6.67 | 0.72 | 5.23 | 4.42 | 4.22 | 7.37 | 3.15 |
| I22 | Stilo, Camini, Riace, Stignano | 7.06 | 0.78 | 5.30 | 4.48 | 4.55 | 7.90 | 3.35 |
| I23 | Caulonia | 6.38 | 0.77 | 5.25 | 4.45 | 4.28 | 7.32 | 3.04 |
| I24 | Roccella Ionica, Marina di Gioiosa Ionica, Grotteria, Siderno | 6.37 | 0.69 | 5.14 | 4.35 | 3.89 | 6.74 | 2.85 |
| I25 | Locri, Portigliola, Sant'Ilario dello Ionio, Ardore | 6.36 | 0.67 | 5.15 | 4.37 | 3.85 | 6.70 | 2.85 |
| I26 | Bovalino, Casignana, Bianco, Africo | 6.62 | 0.72 | 5.26 | 4.46 | 4.15 | 7.24 | 3.09 |
| I27 | Ferruzzano, Bruzzano Zeffirio | 7.37 | 0.76 | 5.33 | 4.52 | 4.35 | 7.45 | 3.10 |
| I28 | Brancaleone | 6.56 | 0.69 | 5.41 | 4.59 | 4.21 | 7.43 | 3.22 |
| I29 | Palizzi | 6.5 | 0.77 | 5.32 | 4.50 | 4.26 | 7.19 | 2.93 |
| I30 | Bova Marina, Condofuri, San Lorenzo | 6.51 | 0.77 | 5.18 | 4.39 | 4.19 | 7.08 | 2.89 |
| I31 | Melito di Porto Salvo, Montebello Ionico | 6.39 | 0.78 | 5.04 | 4.28 | 4.02 | 6.71 | 2.69 |
| I32 | Motta San Giovanni, Reggio Calabria | 6.35 | 0.68 | 4.80 | 4.10 | 3.64 | 6.21 | 2.57 |
| T1 | Villa San Giovanni, Scilla, Bagnara Calabra | 6.71 | 0.48 | 5.04 | 4.20 | 3.61 | 6.95 | 3.34 |
| T2 | Seminara, Palmi | 8.29 | 0.62 | 5.14 | 4.31 | 4.42 | 8.06 | 3.64 |
| T3 | Gioia Tauro, San Ferdinando, Rosarno | 8.14 | 0.63 | 5.11 | 4.27 | 4.39 | 7.92 | 3.53 |
| T4 | Nicotera, Joppolo, Ricadi up to Capo Vaticano | 7.81 | 0.52 | 5.29 | 4.38 | 3.89 | 7.18 | 3.29 |
| T5 | Ricadi from Capo Vaticano to the north | 9.4 | 0.74 | 5.38 | 4.49 | 4.90 | 8.89 | 3.99 |
| T6 | Tropea, Parghelia, Zambrone | 9.22 | 0.69 | 5.46 | 4.55 | 4.74 | 8.76 | 4.02 |
| T7 | Briatico, Vibo Marina, Pizzo, Curinga | 9.03 | 0.58 | 5.51 | 4.59 | 4.25 | 8.15 | 3.90 |

**Table 1.** *Cont.*

| Point | Coastal Municipality | $h_{s,max}$ [m] | $h_s$ [m] | $t_p$ [s] | $t_m$ [s] | $h_{s1}$ [m] | $h_{s100}$ [m] | $\Delta h_{s1-100}$ [m] |
|---|---|---|---|---|---|---|---|---|
| T8 | Lamezia Terme, Gizzeria | 8.54 | 0.59 | 5.50 | 4.60 | 4.20 | 7.92 | 3.72 |
| T9 | Falerna, Nocera Terinese | 9.82 | 0.75 | 5.50 | 4.63 | 4.98 | 9.18 | 4.20 |
| T10 | Amantea | 9.58 | 0.73 | 5.51 | 4.65 | 4.82 | 8.79 | 3.97 |
| T11 | Belmonte Calabro, Longobardi, Fiumefreddo Bruzio | 9.44 | 0.73 | 5.53 | 4.68 | 4.75 | 8.65 | 3.90 |
| T12 | Falconara Albanese, San Lucido, Paola to the San Domenico River mouth | 9.16 | 0.71 | 5.55 | 4.70 | 4.64 | 8.46 | 3.82 |
| T13 | Paola from the San Domenico River mouth to the north, Fuscaldo, Guardia Piemontese | 8.31 | 0.67 | 5.54 | 4.66 | 4.29 | 7.75 | 3.46 |
| T14 | Acquappesa, Cetraro | 9.49 | 0.80 | 5.57 | 4.70 | 4.93 | 8.77 | 3.84 |
| T15 | Bonifati, Sangineto, Belvedere Marittimo to the Di Mare River mouth | 9.45 | 0.80 | 5.55 | 4.66 | 4.90 | 8.54 | 3.64 |
| T16 | Belvedere Marittimo from Di Mare River mouth, Diamante to the port | 9.47 | 0.78 | 5.64 | 4.75 | 4.91 | 8.81 | 3.90 |
| T17 | Diamante from the port to the north, Grisolia, Santa Maria del Cedro | 8.86 | 0.73 | 5.66 | 4.76 | 4.66 | 8.32 | 3.66 |
| T18 | Scalea | 8.3 | 0.69 | 5.66 | 4.76 | 4.49 | 7.99 | 3.50 |
| T19 | San Nicola Arcella, Praia a Mare, Tortora | 7.15 | 0.63 | 5.60 | 4.70 | 4.12 | 7.24 | 3.12 |
| | Maximum | 9.82 | 0.92 | 5.66 | 4.76 | 4.98 | 9.18 | 4.20 |
| | Minimum | 4.91 | 0.43 | 3.94 | 3.36 | 2.76 | 4.71 | 1.95 |
| | Average | 7.33 | 0.70 | 5.13 | 4.33 | 4.23 | 7.43 | 3.20 |

**Table 2.** Values of average annual energy flux $\Phi_t$, main *MS*, secondary *SS*, and tertiary *TS* sector and corresponding values of energy flux $\Phi_{MS}$, $\Phi_{SS}$, $\Phi_{TS}$, ratio between the main sector energy flux value and the total energy flux value $\Phi_{MS}/\Phi_t$, and characteristic parameters $u$, $w$, $a_{10}$, $b_{10}$. Legend: the higher values are in red, and the lower values are in green.

| Point | $\Phi_t$ [kW/m] | MS [°] | $\Phi_{MS}$ [kW/m] | SS [°] | $\Phi_{SS}$ [kW/m] | TS [°] | $\Phi_{TS}$ [kW/m] | $\Phi_{MS}/\Phi_t$ [%] | $u$ | $w$ [m] | $a_{10}$ [m] | $b_{10}$ [h] |
|---|---|---|---|---|---|---|---|---|---|---|---|---|
| I1 | 3.5 | 130 | 1.01 | | | | | 29 | 0.94 | 0.47 | 2.83 | 66.65 |
| I2 | 3.7 | 130 | 1.15 | | | | | 31 | 0.98 | 0.51 | 2.89 | 65.77 |
| I3 | 3.8 | 130 | 0.86 | 230 | 0.34 | | | 23 | 1.05 | 0.58 | 2.92 | 63.40 |
| I4 | 2.5 | 110 | 0.55 | 240 | 0.24 | | | 22 | 1.03 | 0.47 | 2.52 | 60.29 |
| I5 | 1.7 | 100 | 0.28 | 250 | 0.19 | 30 | 0.15 | 17 | 1.02 | 0.40 | 2.22 | 58.48 |
| I6 | 3.0 | 120 | 0.36 | 0 | 0.27 | 280 | 0.21 | 12 | 1.04 | 0.52 | 2.79 | 59.25 |
| I7 | 3.4 | 0 | 0.44 | 110 | 0.36 | 300 | 0.15 | 13 | 1.06 | 0.56 | 2.92 | 66.18 |
| I8 | 5.4 | 130 | 0.87 | 350 | 0.54 | | | 16 | 1.09 | 0.70 | 3.37 | 73.26 |
| I9 | 7.1 | 140 | 0.95 | 350 | 0.76 | | | 13 | 1.09 | 0.77 | 3.70 | 82.51 |
| I10 | 7.2 | 140 | 0.94 | 350 | 0.72 | | | 13 | 1.10 | 0.78 | 3.71 | 82.76 |
| I11 | 7.4 | 140 | 0.90 | 0 | 0.81 | | | 12 | 1.09 | 0.79 | 3.73 | 85.06 |
| I12 | 7.6 | 140 | 0.86 | 0 | 0.82 | | | 11 | 1.10 | 0.79 | 3.74 | 85.30 |
| I13 | 8.0 | 140 | 0.75 | 10 | 0.76 | 190 | 0.42 | 9 | 1.11 | 0.82 | 3.77 | 84.97 |
| I14 | 9.2 | 10 | 0.86 | 140 | 0.75 | 200 | 0.55 | 9 | 1.15 | 0.90 | 3.93 | 85.96 |
| I15 | 7.2 | 140 | 0.69 | 190 | 0.51 | 270 | 0.34 | 10 | 1.09 | 0.76 | 3.53 | 83.75 |
| I16 | 6.2 | 140 | 0.66 | 190 | 0.43 | 270 | 0.32 | 11 | 1.08 | 0.70 | 3.32 | 81.35 |
| I17 | 5.1 | 130 | 0.73 | | | | | 14 | 1.01 | 0.60 | 3.08 | 84.83 |
| I18 | 5.0 | 130 | 0.74 | | | | | 15 | 0.97 | 0.55 | 3.01 | 86.89 |
| I19 | 5.2 | 130 | 0.66 | 50 | 0.18 | | | 13 | 1.00 | 0.58 | 3.07 | 82.13 |
| I20 | 5.9 | 130 | 0.70 | 50 | 0.34 | | | 12 | 1.00 | 0.62 | 3.27 | 83.23 |
| I21 | 6.2 | 130 | 0.64 | 50 | 0.46 | | | 10 | 1.03 | 0.66 | 3.35 | 81.77 |
| I22 | 7.3 | 130 | 0.70 | 50 | 0.62 | 200 | 0.31 | 10 | 1.04 | 0.72 | 3.61 | 83.61 |
| I23 | 6.8 | 130 | 0.68 | 50 | 0.36 | | | 10 | 1.07 | 0.72 | 3.42 | 84.61 |
| I24 | 5.5 | 130 | 0.71 | | | | | 13 | 1.05 | 0.64 | 3.09 | 89.43 |
| I25 | 5.2 | 120 | 0.64 | 70 | 0.19 | | | 12 | 1.04 | 0.62 | 3.05 | 86.91 |
| I26 | 6.0 | 130 | 0.58 | 60 | 0.40 | | | 10 | 1.03 | 0.65 | 3.26 | 82.76 |
| I27 | 6.9 | 120 | 0.61 | 60 | 0.54 | | | 9 | 1.06 | 0.71 | 3.48 | 82.77 |
| I28 | 6.2 | 120 | 0.71 | 60 | 0.37 | | | 11 | 1.02 | 0.65 | 3.29 | 87.68 |
| I29 | 6.8 | 120 | 0.58 | 60 | 0.34 | 300 | 0.17 | 9 | 1.09 | 0.72 | 3.40 | 82.19 |
| I30 | 6.4 | 120 | 0.60 | 320 | 0.26 | | | 9 | 1.08 | 0.70 | 3.33 | 78.38 |
| I31 | 6.0 | 120 | 0.60 | 330 | 0.38 | | | 10 | 1.10 | 0.70 | 3.22 | 78.01 |
| I32 | 4.4 | 160 | 0.47 | 340 | 0.20 | | | 11 | 1.05 | 0.57 | 2.91 | 70.10 |
| T1 | 3.1 | 310 | 0.74 | | | | | 24 | 0.87 | 0.40 | 2.76 | 66.79 |
| T2 | 5.2 | 300 | 1.18 | | | | | 23 | 0.94 | 0.57 | 3.48 | 69.86 |
| T3 | 5.3 | 300 | 0.97 | | | | | 18 | 0.96 | 0.59 | 3.47 | 69.42 |
| T4 | 3.9 | 280 | 1.11 | | | | | 28 | 0.93 | 0.48 | 3.10 | 68.04 |
| T5 | 7.4 | 280 | 1.53 | | | | | 21 | 0.96 | 0.67 | 3.90 | 76.96 |
| T6 | 6.8 | 290 | 1.50 | | | | | 22 | 0.94 | 0.62 | 3.76 | 76.97 |
| T7 | 5.1 | 290 | 1.55 | | | | | 30 | 0.89 | 0.51 | 3.36 | 78.57 |
| T8 | 5.1 | 270 | 1.59 | | | | | 31 | 0.91 | 0.51 | 3.32 | 75.96 |
| T9 | 8.0 | 280 | 2.05 | | | | | 26 | 0.95 | 0.67 | 3.96 | 83.01 |
| T10 | 7.6 | 270 | 2.01 | | | | | 26 | 0.96 | 0.67 | 3.84 | 83.20 |

**Table 2.** *Cont.*

| Point | $\Phi_t$ [kW/m] | MS [°] | $\Phi_{MS}$ [kW/m] | SS [°] | $\Phi_{SS}$ [kW/m] | TS [°] | $\Phi_{TS}$ [kW/m] | $\Phi_{MS}/\Phi_t$ [%] | $u$ | $w$ [m] | $a_{10}$ [m] | $b_{10}$ [h] |
|---|---|---|---|---|---|---|---|---|---|---|---|---|
| T11 | 7.5 | 270 | 2.11 | | | | | 28 | 0.97 | 0.67 | 3.78 | 84.16 |
| T12 | 7.2 | 270 | 2.08 | | | | | 29 | 0.97 | 0.65 | 3.68 | 84.17 |
| T13 | 6.2 | 260 | 2.01 | | | | | 33 | 0.98 | 0.62 | 3.40 | 83.72 |
| T14 | 8.4 | 260 | 2.21 | | | | | 26 | 1.00 | 0.73 | 3.89 | 83.66 |
| T15 | 8.4 | 260 | 1.84 | | | | | 22 | 1.03 | 0.75 | 3.88 | 78.96 |
| T16 | 8.4 | 260 | <span style="color:red">2.39</span> | | | | | 28 | 0.99 | 0.72 | 3.89 | 84.66 |
| T17 | 7.4 | 260 | 2.23 | | | | | 30 | 0.99 | 0.68 | 3.68 | 81.41 |
| T18 | 6.8 | 260 | 2.08 | | | | | 31 | 0.99 | 0.65 | 3.54 | 79.06 |
| T19 | 5.5 | 250 | 1.88 | | | | | <span style="color:red">34</span> | 1.00 | 0.60 | 3.25 | 73.63 |
| Max | 9.2 | | 2.39 | | 0.82 | | 0.55 | 34 | 1.15 | 0.90 | 3.96 | 89.43 |
| Min. | 1.7 | | 0.28 | | 0.18 | | 0.15 | 9 | 0.87 | 0.40 | 2.22 | 58.48 |
| Av. | 6.0 | | 1.09 | | 0.45 | | 0.29 | 18 | 1.02 | 0.64 | 3.37 | 78.28 |

**Table 3.** Values of seasonal maximum significant wave height $h_{s,max,i}$ *and of* seasonal average significant wave height $h_{s,i}$, where the i-th season has been indicated with subscripts *W* for winter, *SP* for spring, *SU* for summer and *A* for autumn. Legend: the higher values are in red, and the lower values are in green.

| Point | $h_{s,max,W}$ [m] | $h_{s,max,SP}$ [m] | $h_{s,max,SU}$ [m] | $h_{s,max,A}$ [m] | $h_{s,W}$ [m] | $h_{s,SP}$ [m] | $h_{s,SU}$ [m] | $h_{s,A}$ [m] |
|---|---|---|---|---|---|---|---|---|
| I1 | 5.68 | 3.55 | 3.19 | 5.66 | 0.72 | 0.48 | 0.34 | 0.67 |
| I2 | 5.72 | 3.47 | 3.36 | 5.72 | 0.76 | 0.50 | 0.37 | 0.70 |
| I3 | 5.55 | 3.37 | 3.43 | 5.88 | 0.79 | 0.52 | 0.39 | 0.71 |
| I4 | 5.23 | 2.77 | 3.28 | 5.57 | 0.65 | 0.43 | 0.33 | 0.58 |
| I5 | <span style="color:green">4.67</span> | <span style="color:green">2.65</span> | 2.99 | <span style="color:green">4.91</span> | <span style="color:green">0.55</span> | <span style="color:green">0.37</span> | <span style="color:green">0.31</span> | <span style="color:green">0.48</span> |
| I6 | 5.87 | 3.57 | 3.46 | 6.04 | 0.71 | 0.45 | 0.37 | 0.64 |
| I7 | 6.13 | 4.02 | 3.43 | 6.18 | 0.74 | 0.46 | 0.39 | 0.67 |
| I8 | 6.26 | 4.23 | 3.79 | 6.87 | 0.93 | 0.59 | 0.47 | 0.86 |
| I9 | 6.77 | 4.4 | 4.2 | 7.42 | 1.05 | 0.64 | 0.50 | 0.98 |
| I10 | 6.8 | 4.27 | 4.32 | 7.44 | 1.05 | 0.63 | 0.49 | 0.99 |
| I11 | 6.83 | 4.29 | 4.42 | 7.34 | 1.06 | 0.64 | 0.49 | 1.00 |
| I12 | 6.78 | 4.24 | 4.48 | 7.23 | 1.08 | 0.65 | 0.50 | 1.02 |
| I13 | 6.66 | 4.29 | 4.53 | 7.02 | 1.12 | 0.68 | 0.52 | 1.05 |
| I14 | 6.71 | 4.33 | 4.79 | 7.04 | <span style="color:red">1.21</span> | <span style="color:red">0.76</span> | <span style="color:red">0.58</span> | <span style="color:red">1.13</span> |
| I15 | 6.54 | 4.01 | 4.72 | 6.69 | 1.06 | 0.72 | 0.53 | 0.99 |
| I16 | 6.34 | 3.77 | 4.64 | 6.45 | 0.97 | 0.68 | 0.49 | 0.91 |
| I17 | 6.24 | 3.63 | 4.67 | 6.28 | 0.86 | 0.61 | 0.44 | 0.80 |
| I18 | 6.11 | 3.67 | 4.69 | 6.03 | 0.85 | 0.60 | 0.44 | 0.80 |
| I19 | 6.11 | 3.71 | 4.63 | 5.84 | 0.88 | 0.60 | 0.44 | 0.83 |
| I20 | 6.51 | 3.88 | 4.86 | 6.18 | 0.93 | 0.60 | 0.44 | 0.88 |
| I21 | 6.67 | 3.98 | 4.7 | 6.04 | 0.96 | 0.60 | 0.44 | 0.91 |
| I22 | 7.06 | 4.27 | <span style="color:red">5.03</span> | 6.28 | 1.05 | 0.64 | 0.46 | 0.99 |
| I23 | 6.38 | 4.33 | 4.75 | 6.31 | 1.01 | 0.66 | 0.46 | 0.95 |
| I24 | 6.37 | 4.16 | 4.59 | 5.92 | 0.88 | 0.61 | 0.43 | 0.84 |
| I25 | 6.36 | 4.12 | 4.33 | 5.8 | 0.86 | 0.60 | 0.42 | 0.82 |
| I26 | 6.62 | 4.22 | 4.35 | 5.97 | 0.94 | 0.61 | 0.44 | 0.89 |
| I27 | 7.37 | 4.34 | 4.42 | 6.17 | 1.01 | 0.63 | 0.45 | 0.96 |
| I28 | 6.56 | 4.41 | 4.34 | 6.09 | 0.95 | 0.58 | 0.36 | 0.90 |
| I29 | 6.4 | 4.35 | 4.23 | 6.5 | 1.02 | 0.67 | 0.45 | 0.96 |
| I30 | 6.41 | 4.23 | 4.01 | 6.51 | 0.99 | 0.69 | 0.49 | 0.93 |
| I31 | 6.39 | 4.03 | 3.64 | 6.19 | 0.97 | 0.71 | 0.54 | 0.90 |
| I32 | 5.91 | 3.65 | 3.32 | 6.35 | 0.82 | 0.65 | 0.50 | 0.76 |
| T1 | 6.71 | 3.53 | <span style="color:green">2.67</span> | 6.61 | 0.65 | 0.41 | 0.31 | 0.57 |
| T2 | 8.15 | 4.8 | 3.49 | 8.29 | 0.84 | 0.53 | 0.40 | 0.73 |
| T3 | 7.91 | 4.84 | 3.59 | 8.14 | 0.84 | 0.54 | 0.41 | 0.75 |
| T4 | 6.74 | 4.76 | 3.51 | 7.81 | 0.69 | 0.44 | 0.33 | 0.61 |
| T5 | 8.73 | 5.69 | 4.16 | 9.4 | 0.99 | 0.63 | 0.48 | 0.87 |

**Table 3.** *Cont.*

| Point | $h_{s,max,W}$ [m] | $h_{s,max,SP}$ [m] | $h_{s,max,SU}$ [m] | $h_{s,max,A}$ [m] | $h_{s,W}$ [m] | $h_{s,SP}$ [m] | $h_{s,SU}$ [m] | $h_{s,A}$ [m] |
|---|---|---|---|---|---|---|---|---|
| T6 | 8.78 | 5.52 | 4.05 | 9.22 | 0.93 | 0.58 | 0.45 | 0.81 |
| T7 | 8.32 | 5.38 | 3.76 | 9.03 | 0.77 | 0.49 | 0.39 | 0.66 |
| T8 | 7.47 | 5.44 | 3.92 | 8.54 | 0.78 | 0.51 | 0.37 | 0.69 |
| T9 | <span style="color:red">9.1</span> | 6.1 | 4.66 | <span style="color:red">9.82</span> | 1.00 | 0.65 | 0.48 | 0.88 |
| T10 | 8.82 | 5.84 | 4.61 | 9.58 | 0.98 | 0.63 | 0.47 | 0.86 |
| T11 | 8.55 | 5.73 | 4.66 | 9.44 | 0.97 | 0.62 | 0.46 | 0.86 |
| T12 | 8.11 | 5.61 | 4.68 | 9.16 | 0.95 | 0.60 | 0.45 | 0.85 |
| T13 | 7.25 | 5.3 | 4.47 | 8.31 | 0.89 | 0.56 | 0.41 | 0.81 |
| T14 | 8.26 | 5.9 | 5 | 9.49 | 1.05 | 0.67 | 0.50 | 0.97 |
| T15 | 8.14 | 5.99 | 4.84 | 9.45 | 1.06 | 0.67 | 0.51 | 0.97 |
| T16 | 8.01 | 6.31 | 4.92 | 9.47 | 1.02 | 0.66 | 0.50 | 0.94 |
| T17 | 7.35 | 6.38 | 4.61 | 8.86 | 0.95 | 0.62 | 0.46 | 0.87 |
| T18 | 7.01 | <span style="color:red">6.48</span> | 4.43 | 8.3 | 0.91 | 0.58 | 0.44 | 0.84 |
| T19 | 6.36 | 5.99 | 4.01 | 7.15 | 0.83 | 0.52 | 0.38 | 0.77 |
| Max | 9.1 | 6.48 | 5.03 | 9.82 | 1.21 | 0.76 | 0.58 | 1.13 |
| Min. | 4.67 | 2.65 | 2.67 | 4.91 | 0.55 | 0.37 | 0.31 | 0.48 |
| Av. | 6.90 | 4.55 | 4.19 | 7.22 | 0.91 | 0.59 | 0.44 | 0.84 |

**Table 4.** Values of seasonal average annual energy flux $\Phi_{t,i}$, and of seasonal main sector $MS_i$, where the i-th season has been indicated with subscripts *W* for winter, *SP* for spring, *SU* for summer and *A* for autumn. Legend: the higher values are in red, and the lower values are in green.

| Point | $\Phi_{t,W}$ [kW/m] | $\Phi_{t,SP}$ [kW/m] | $\Phi_{t,SU}$ [kW/m] | $\Phi_{t,A}$ [kW/m] | $\Phi_{t,W}$ [%] | $\Phi_{t,SP}$ [%] | $\Phi_{t,SU}$ [%] | $\Phi_{t,A}$ [%] | $MS_W$ [°] | $MS_{SP}$ [°] | $MS_{SU}$ [°] | $MS_A$ [°] |
|---|---|---|---|---|---|---|---|---|---|---|---|---|
| I1 | 1.47 | 0.52 | 0.21 | 1.33 | 0.42 | 0.15 | 0.06 | <span style="color:red">0.38</span> | 130 | 140 | 140 | 130 |
| I2 | 1.56 | 0.55 | 0.24 | 1.38 | 0.42 | 0.15 | 0.06 | 0.37 | 130 | 130 | 130 | 130 |
| I3 | 1.59 | 0.57 | 0.28 | 1.37 | 0.42 | 0.15 | 0.07 | 0.36 | 130 | 130 | 230 | 120 |
| I4 | 1.05 | 0.36 | 0.20 | 0.90 | 0.42 | 0.14 | 0.08 | 0.36 | 110 | 110 | 240 | 110 |
| I5 | <span style="color:green">0.71</span> | <span style="color:green">0.24</span> | <span style="color:green">0.16</span> | <span style="color:green">0.57</span> | 0.42 | 0.14 | <span style="color:red">0.09</span> | <span style="color:green">0.33</span> | 100 | 250 | 250 | 100 |
| I6 | 1.29 | 0.41 | 0.24 | 1.09 | 0.43 | 0.13 | 0.08 | 0.36 | 120 | 280 | 280 | 110 |
| I7 | 1.46 | 0.42 | 0.27 | 1.23 | 0.43 | <span style="color:green">0.12</span> | 0.08 | 0.36 | 350 | 0 | 0 | 110 |
| I8 | 2.29 | 0.73 | 0.41 | 2.00 | 0.42 | 0.13 | 0.08 | 0.37 | 130 | 140 | 350 | 130 |
| I9 | 3.02 | 0.93 | 0.48 | 2.70 | 0.42 | 0.13 | 0.07 | <span style="color:red">0.38</span> | 140 | 140 | 350 | 130 |
| I10 | 3.06 | 0.93 | 0.47 | 2.74 | 0.43 | 0.13 | 0.06 | <span style="color:red">0.38</span> | 140 | 140 | 0 | 130 |
| I11 | 3.16 | 0.96 | 0.48 | 2.82 | 0.43 | 0.13 | 0.06 | <span style="color:red">0.38</span> | 140 | 130 | 0 | 130 |
| I12 | 3.25 | 1.00 | 0.49 | 2.91 | 0.42 | 0.13 | 0.06 | <span style="color:red">0.38</span> | 140 | 140 | 0 | 130 |
| I13 | 3.39 | 1.07 | 0.53 | 3.03 | 0.42 | 0.13 | 0.07 | <span style="color:red">0.38</span> | 140 | 150 | 10 | 130 |
| I14 | <span style="color:red">3.87</span> | 1.27 | 0.65 | <span style="color:red">3.44</span> | 0.42 | 0.14 | 0.07 | 0.37 | 140 | 150 | 10 | 10 |
| I15 | 2.92 | 1.13 | 0.51 | 2.62 | 0.41 | 0.16 | 0.07 | 0.37 | 140 | 150 | 270 | 130 |
| I16 | 2.51 | 0.99 | 0.43 | 2.25 | 0.41 | 0.16 | 0.07 | 0.36 | 140 | 150 | 270 | 130 |
| I17 | 2.10 | 0.81 | 0.33 | 1.87 | 0.41 | 0.16 | 0.06 | 0.37 | 130 | 140 | 280 | 130 |
| I18 | 2.04 | 0.79 | 0.33 | 1.80 | 0.41 | 0.16 | 0.07 | 0.36 | 130 | 140 | 280 | 130 |
| I19 | 2.15 | 0.80 | 0.34 | 1.90 | 0.41 | 0.15 | 0.07 | 0.37 | 130 | 140 | 300 | 130 |
| I20 | 2.47 | 0.85 | 0.36 | 2.18 | 0.42 | 0.14 | 0.06 | 0.37 | 130 | 140 | 310 | 120 |
| I21 | 2.61 | 0.88 | 0.36 | 2.31 | 0.42 | 0.14 | 0.06 | <span style="color:red">0.38</span> | 130 | 140 | 50 | 120 |
| I22 | 3.12 | 1.01 | 0.42 | 2.76 | 0.43 | 0.14 | 0.06 | <span style="color:red">0.38</span> | 130 | 140 | 50 | 120 |
| I23 | 2.84 | 1.04 | 0.41 | 2.50 | 0.42 | 0.15 | 0.06 | 0.37 | 130 | 140 | 60 | 120 |
| I24 | 2.27 | 0.89 | 0.34 | 1.98 | 0.41 | 0.16 | 0.06 | 0.36 | 130 | 140 | 310 | 120 |
| I25 | 2.18 | 0.85 | 0.33 | 1.88 | 0.42 | 0.16 | 0.06 | 0.36 | 130 | 130 | 310 | 120 |
| I26 | 2.54 | 0.92 | 0.37 | 2.20 | 0.42 | 0.15 | 0.06 | 0.37 | 130 | 130 | 330 | 120 |
| I27 | 2.93 | 1.00 | 0.40 | 2.54 | 0.43 | 0.15 | 0.06 | 0.37 | 120 | 130 | 60 | 120 |
| I28 | 2.67 | 0.93 | 0.30 | 2.30 | 0.43 | 0.15 | <span style="color:green">0.05</span> | 0.37 | 120 | 130 | 70 | 110 |
| I29 | 2.84 | 1.08 | 0.41 | 2.43 | 0.42 | 0.16 | 0.06 | 0.36 | 120 | 130 | 300 | 110 |
| I30 | 2.62 | 1.10 | 0.45 | 2.22 | 0.41 | 0.17 | 0.07 | 0.35 | 120 | 130 | 320 | 110 |
| I31 | 2.41 | 1.07 | 0.49 | 2.01 | 0.40 | 0.18 | 0.08 | 0.34 | 120 | 130 | 330 | 110 |
| I32 | 1.71 | 0.84 | 0.39 | 1.43 | <span style="color:green">0.39</span> | <span style="color:red">0.19</span> | <span style="color:red">0.09</span> | <span style="color:green">0.33</span> | 130 | 150 | 340 | 120 |
| T1 | 1.34 | 0.46 | 0.26 | 1.01 | <span style="color:red">0.44</span> | 0.15 | 0.08 | <span style="color:green">0.33</span> | 310 | 310 | 310 | 310 |
| T2 | 2.26 | 0.80 | 0.45 | 1.72 | 0.43 | 0.15 | <span style="color:red">0.09</span> | <span style="color:green">0.33</span> | 290 | 300 | 310 | 300 |
| T3 | 2.26 | 0.82 | 0.46 | 1.77 | 0.43 | 0.15 | <span style="color:red">0.09</span> | <span style="color:green">0.33</span> | 290 | 290 | 300 | 300 |
| T4 | 1.68 | 0.62 | 0.33 | 1.30 | 0.43 | 0.16 | <span style="color:red">0.08</span> | <span style="color:green">0.33</span> | 280 | 280 | 290 | 280 |
| T5 | 3.16 | 1.15 | 0.64 | 2.46 | 0.43 | 0.15 | <span style="color:red">0.09</span> | <span style="color:green">0.33</span> | 280 | 280 | 300 | 280 |
| T6 | 2.92 | 1.02 | 0.59 | 2.25 | 0.43 | 0.15 | <span style="color:red">0.09</span> | <span style="color:green">0.33</span> | 290 | 290 | 300 | 280 |
| T7 | 2.19 | 0.80 | 0.45 | 1.69 | 0.43 | 0.16 | <span style="color:red">0.09</span> | <span style="color:green">0.33</span> | 280 | 280 | 290 | 280 |
| T8 | 2.16 | 0.81 | 0.42 | 1.69 | 0.43 | 0.16 | 0.08 | <span style="color:green">0.33</span> | 270 | 280 | 280 | 270 |
| T9 | 3.36 | 1.27 | 0.68 | 2.64 | 0.42 | 0.16 | <span style="color:red">0.09</span> | <span style="color:green">0.33</span> | 270 | 280 | 290 | 270 |
| T10 | 3.20 | 1.21 | 0.64 | 2.53 | 0.42 | 0.16 | 0.08 | <span style="color:green">0.33</span> | 270 | 270 | 290 | 270 |

**Table 4.** *Cont.*

| Point | $\Phi_{t,W}$ [kW/m] | $\Phi_{t,SP}$ [kW/m] | $\Phi_{t,SU}$ [kW/m] | $\Phi_{t,A}$ [kW/m] | $\Phi_{t,W}$ [%] | $\Phi_{t,SP}$ [%] | $\Phi_{t,SU}$ [%] | $\Phi_{t,A}$ [%] | $MS_W$ [°] | $MS_{SP}$ [°] | $MS_{SU}$ [°] | $MS_A$ [°] |
|---|---|---|---|---|---|---|---|---|---|---|---|---|
| T11 | 3.14 | 1.19 | 0.63 | 2.50 | 0.42 | 0.16 | 0.08 | 0.33 | 270 | 270 | 280 | 270 |
| T12 | 3.00 | 1.14 | 0.60 | 2.42 | 0.42 | 0.16 | 0.08 | 0.34 | 270 | 270 | 280 | 270 |
| T13 | 2.58 | 0.98 | 0.49 | 2.12 | 0.42 | 0.16 | 0.08 | 0.34 | 260 | 260 | 270 | 260 |
| T14 | 3.49 | 1.35 | 0.70 | 2.91 | 0.41 | 0.16 | 0.08 | 0.34 | 260 | 270 | 280 | 260 |
| T15 | 3.45 | 1.34 | 0.71 | 2.90 | 0.41 | 0.16 | 0.08 | 0.35 | 260 | 270 | 280 | 260 |
| T16 | 3.44 | 1.36 | 0.71 | 2.89 | 0.41 | 0.16 | 0.08 | 0.34 | 260 | 260 | 280 | 260 |
| T17 | 3.03 | 1.20 | 0.61 | 2.57 | 0.41 | 0.16 | 0.08 | 0.35 | 260 | 260 | 270 | 260 |
| T18 | 2.76 | 1.09 | 0.54 | 2.36 | 0.41 | 0.16 | 0.08 | 0.35 | 260 | 260 | 260 | 260 |
| T19 | 2.23 | 0.87 | 0.41 | 1.94 | 0.41 | 0.16 | 0.08 | 0.36 | 250 | 250 | 250 | 250 |
| Max | 3.87 | 1.36 | 0.71 | 3.44 | 0.44 | 0.19 | 0.09 | 0.38 | | | | |
| Min. | 0.71 | 0.24 | 0.16 | 0.57 | 0.39 | 0.12 | 0.05 | 0.33 | | | | |
| Av. | 2.51 | 0.91 | 0.44 | 2.13 | 0.4 | 0.2 | 0.1 | 0.4 | | | | |

The $h_s$ values vary between 0.43 m at point I5, and 0.92 m at point I14. The average value is 0.7 m and this parameter is characterized by an increasing and decreasing trend between the various points in agreement with that observed with $h_{s,max}$. However, unlike this, the $h_s$ values obtained in the two macro-areas are similar and the value maximum is observed in the Ionian coast. Another difference with $h_{s,max}$ concerns the trends between Punta Alice and Capo Rizzuto (points from I9 to I14) and between points I28 and I32. Indeed, in both cases, $h_{s,max}$ decreases while $h_s$ grows.

The $t_p$ values vary between 3.94 s at point I5, and 5.66 s at points T17 and T18, located in the northern Tyrrhenian coast. The average value is 5.13 s, and the variations between the various points are like those of $h_{s,max}$. However, the maximum $t_p$ values obtained in the two macro areas are not very different from each other and are equal to 5.41 s in the Ionian coast (point I28, in the southern part) and 5.66 s in the Tyrrhenian coast.

The $t_m$ values vary between 3.36 s at point I5, and 4.76 s at points T17 and T18. The average value is 4.33 s, and the variations between the various points are like those of $t_p$.

The $h_{s1}$ values vary between 2.76 m at point I5, and 4.98 m at point T9, located in the central-northern Tyrrhenian Sea. The average value is 4.23 m. Additionally, this parameter has a variability between the various points like that of $h_{s,max}$. However, the maximum values obtained in the two macro areas are not very different from each other and are equal to 4.77 m in the Ionian coast (point I14) and to 4.98 m in the Tyrrhenian coast.

The values of $h_{s100}$ vary between 4.71 m at point I5, and 9.18 m at point T9. The average value is 7.43 m, and the variations between the various points are like those of $h_{s1}$. However, unlike what was observed with $h_{s1}$, the $h_{s100}$ values obtained in the two macro areas are very different from each other. In fact, the maximum $h_{s100}$ value in the Ionian coast is 7.9 m (at point I22, in the central-southern part), while in the Tyrrhenian coast, in 12 points out of 19, the value of $h_{s100}$ largely exceeds 8 m, and in one point, it even exceeds 9 m. Consequently, the differences between these values $\Delta h_{s1-100}$ vary between 1.95 m at point I5, and 4.2 m at point T9. The average value is 3.2 m, and the variations between the various points are like those of $h_{s100}$, where the Tyrrhenian coast are characterized by generally higher values than the Ionian coast.

The $\Phi_t$ values vary from just over 1.70 kW/m at point I5 to over 9.2 kW/m at point I14. The average value is about 6.0 kW/m, and this parameter is characterized by an increasing and decreasing trend between the various points in agreement with that observed with $h_{s,max}$. However, unlike this, in many cases, the $\Phi_t$ values reached in the two macro-areas Ionic and Tyrrhenian are similar, and the maximum value is observed in the Ionian coast. Another difference with $h_{s,max}$ concerns the trend between Punta Alice and Capo Rizzuto (points from I9 to I14), where $h_{s,max}$ decreases while $\Phi_t$ grows.

Regarding the main sectors $MS$, in the Ionian coast, they generally come from the south-east directions, from sectors between 100 and 140° to the north. Exceptions are point I32, at the southern mouth of the Strait of Messina, where the main sector $MS$ comes from 160° to the north, and points I7, near the mouth of the Gulf of Taranto, and I14, both characterized from main sectors $MS$ coming from the north direction.

The $\Phi_{MS}$ values vary between just under 0.30 kW/m at point I5, to about 2.4 kW/m at point T16, located in the northern Tyrrhenian Sea. The average value is about 1.10 kW/m; the higher values are in the Tyrrhenian coast, and the variations of its values between the various points are very different from those of the previous parameters. In fact, in the Ionian coast, the $\Phi_{MS}$ values are always lower than 1.00 kW/m, except for points I1 and I2 located in the northern part of the Gulf of Taranto, and only at points I9, I10 and I11, all located near Punta Alice, do the $\Phi_{MS}$ values exceed 0.90 kW/m. Instead, in the Tyrrhenian coast the $\Phi_{MS}$ values are always higher than 1.00 kW/m, and often, they are higher than 2.00 kW/m, except for points T1, located at the northern mouth of the Strait of Messina, and T3, located in the Gulf of Gioia Tauro.

Regarding the secondary sectors *SS*, they are totally absent in the Tyrrhenian coast, while they are almost always present in the Ionian coast, except for points I1 and I2, I17 and I18, located in the Gulf of Squillace, and I24, located in the southern part of the Ionian Sea. The secondary sectors *SS* are characterized by a considerable variability of directions. Most of them come from the east or the north directions, while the others come from the south, south-west, south-east, and north-west directions.

The $\Phi_{SS}$ values vary between just under 0.20 kW/m at point I19, located in the central part of the Ionian coast, to over 0.80 kW/m at point I12, located between Punta Alice and Capo Rizzuto. The average value is 0.45 kW/m.

Regarding the tertiary sectors *TS*, they are present in just nine points, almost all located at the southern mouth of the Gulf of Taranto and around Capo Rizzuto.

The $\Phi_{TS}$ values vary between just under 150 N/s at point I5, and 550 N/s at point I14.

The values of the ratio $\Phi_{MS}/\Phi_t$ vary between 9% at points I13 and I14, both located near Capo Rizzuto, I27, I29 and I30, all in the southern part of the Ionian Sea, and 34% at point T19, located in the Gulf of Policastro. The average value is 18%, and significant differences are observed between the two macro areas. In fact, in the Ionian coast, the $\Phi_{MS}/\Phi_t$ values are always between 9 and 15%, except for the points in the Gulf of Taranto, which reach up to about 30%. On the other hand, in the Tyrrhenian coast, the $\Phi_{MS}/\Phi_t$ values are always higher than 20%, except for point T3.

Regarding the characteristic parameters, it is observed that $u$ varies between 0.87 at point T1, and 1.15 at point I14. The average value is 1.02, and the values observed in the Tyrrhenian coast are generally lower than those in the Ionian coast. The parameter $w$ varies between 0.4 m at point T1, and 0.9 at point I14. The average value is 0.64, and the highest values are observed between Punta Alice and Capo Rizzuto. Parameter $a_{10}$ varies between just over 2.2 m at point I5, and about 4 m at point T9. The average value is about 3.4 m, the lower values are observed in the Gulf of Taranto, and no substantial differences are observed between the two macro areas. The parameter $b_{10}$ varies between just under 60 h at point I5, and about 90 h at point I24. The average value is about 80 h and the variations between the various points are very similar to those of $a_{10}$.

The seasonal analysis highlighted significant differences between the various seasons, the main one being that, in 36 sample areas out of 51, the $h_{s,max,i}$ are not reached in winter but in autumn. These areas are all those of the Tyrrhenian Sea, except point T1, and those of the Ionian Sea between points I3, located in the Gulf of Taranto, and I17 and in the points I29, I30 and I32. Furthermore, it is observed that the maximum $h_{s,max,W}$ values vary between 9.1 m at point T9 and 4.67 m at point I5, with an average value of 6.9 m. The values observed in the Tyrrhenian coast are generally higher than those of the Ionian coast. In fact, in the Ionian coast the maximum $h_{s,max,W}$ exceeds 7 m only at point I27 m, while in the Tyrrhenian coast, the maximum $h_{s,max,W}$ is always higher than 7 m, except at points T1 and T19. The maximum $h_{s,max,SP}$ values vary between 6.48 m at point T18 and 2.65 m at point I5, with an average value of 4.55 m. The values observed in the Tyrrhenian coast are always greater than those of the Ionian coast, except at point T1. The maximum $h_{s,max,SU}$ values vary between 5.03 m at point I22 and 2.67 m at point T1, with an average value of 4.19 m. In this case, the values observed in the Ionian coast are often similar to or higher than the Tyrrhenian one, except in the Gulf of Taranto. The maximum $h_{s,max,A}$ values vary between

9.82 m at point T9 and 4.91 m at point I5, with an average value of 7.22, and have a trend similar to that observed in the maximum $h_{s,max}$. Instead, the $h_{s,i}$ values vary spatially in a homogeneous way between the various seasons, similar to $h_s$ values. The $h_{s,W}$ values vary between 1.21 m and 0.55 m, with an average value 0.91 m. The $h_{s,SP}$ values vary between 0.76 m and 0.37 m, with an average value of 0.59 m. The $h_{s,SU}$ values vary between 0.58 m and 0.31 m, with an average value of 0.44 m. The $h_{s,A}$ values vary between 1.13 m and 0.48 m, with an average value of 0.84 m. In all these cases, the maximum values are observed at point I14, and the minimum values are observed at point I5. From an energy flux point of view, most of the average seasonal flux $\Phi_{t,i}$ is observed in winter, between 39 and 44% of the average annual one, followed by autumn, where there is between 33 and 38% of the average annual one. On the other hand, in the spring, an average flux between 12 and 19% of the average annual one is observed, while in the summer, an average flux between 5 and 9% of the average annual one is observed. In all these cases, there are no significant differences between the Ionian and Tyrrhenian coasts. The minimum seasonal average $\Phi_{t,i}$ values are always observed at point I5, similarly to the average annual flux. Instead, the winter and autumn seasonal maximum $\Phi_{t,i}$ values are always observed at point I9, similarly to the average annual flux, while the spring and summer maximum $\Phi_{t,i}$ values are observed at points T16 and T15, respectively. Finally, regarding the seasonal main sectors $MS_i$, in the Tyrrhenian coast the $MS_i$ are the same as the $MS$ ones, with differences of only a few degrees. Instead, in the Ionian coast, only the main winter, spring and autumnal sectors are the same as the $MS$ ones or differ by a few degrees, except at points I5 and I6 in spring and I7 and I14 in autumn. On the other hand, the main summer sectors often considerably differ from the $MS$ ones. In fact, the $MS$ always come from the south-east while the summer ones come from a range of directions between west and north-east.

## 4. Discussion

In the Results section, the values of each analyzed parameters were described in detail, together with their spatial variations between the various points, but each parameter was analyzed individually. Instead, in this section, a cross analysis of the analyzed parameters will be carried out, also considering the Calabrian peculiarities. Indeed, Calabria represents an interesting case study due to its climatic and geomorphological peculiarities and due to the anthropic action that has affected many coastal and river areas. Indeed, in the second half of the last century, after the end of the Second World War, many of the Calabrian inhabited centers considerably expanded, especially near the sea often in place of beaches and coastal dunes, and especially in the Northern Tyrrhenian coast, between the points T9 and T19. The main effects of these anthropogenic pressures were the triggering of intense coastal erosion processes, the retreats at the mouths of many rivers due to reduced river sediment transport caused by anthropogenic action, and the construction of numerous coastal defenses works. [46,50,71–76]. The anthropization process is more present on the Tyrrhenian coast than on the Ionian one. This can be related to the morphological peculiarities of these coasts. Indeed, the northern Calabrian Tyrrhenian coast, between the points T9 and T19, is characterized by a mountainous relief very close to the coast with few flat coastal areas. Therefore, the inhabited centers have expanded close to the coast. On the other hand, on the Ionian coast, there is generally a greater distance between the coast and the reliefs, so several inhabited centers have been built away from the coast, often behind the existing dunes. Furthermore, many infrastructures and archaeological sites are located a short distance from the coastline and are exposed to the wave action with frequent damaging events [77–79]. Among the main events, it is highlighted that which occurred between the end of October and the beginning of November 2015, where a TLC caused heavy rains and intense sea storms in the central-southern Ionian Calabria, causing the collapse of the road and railway bridges located near the mouth of the Bruzzano River near the point I28 [59,80]. The action of the sea storms also caused damage to the archaeological site of Kaulon during the winter of 2013–2014, in which a part of the dune where the archaeological site is located collapsed [49,81]. Finally, the case of Scilla should

be mentioned. This is a pocket beach with an inhabited center on the back, located at the northern mouth of the Strait of Messina near point T1 and subject to frequent sea storms that flood part of the inhabited center almost annually [24].

One of the main differences between Ionian and Tyrrhenian macro-areas concerns the considerable spatial variability of the various parameters within the Ionian area, while the Tyrrhenian area is more homogeneous, with significant spatial variations of the various parameters only within the gulfs and near the north mouth of the Strait of Messina. These differences can be related to the differences in morphology, exposure to atmospheric disturbances and fetch lengths between the two macro-areas. In fact, the morphology of the Tyrrhenian coasts is generally linear, with gulfs of modest size and with only the promontory of Capo Vaticano to interrupt the linearity of the coast. Furthermore, the Tyrrhenian coasts are exposed to atmospheric disturbances coming almost only from the north-west directions, acting on fetches of lengths always of the order of hundreds of kilometers. On the other hand, the Ionian coasts have an irregular morphology, with very extensive gulfs, especially that of Taranto, which involve very variable fetch lengths. In addition, atmospheric disturbances do not come from a single direction but from varying directions, mainly between north-east and south-east.

Another relevant difference between the two coasts is observed in the values of maximum significant wave height $h_{s,max}$ (Figure 3), of average significant wave height $h_s$ (Figure 4), of significant wave height of return period of 1 year $h_{s1}$ (Figure 5), and of significant wave height of return period of 100 years $h_{s100}$ (Figure 6), where the last two parameters are associated with frequent and extreme events, respectively. In detail, the greatest values of $h_{s,max}$ are observed in the Tyrrhenian coast. A similar situation is also observed for $h_{s100}$, corresponding to extreme events. However, these differences are not observed in terms of $h_s$ and $h_{s1}$, corresponding to frequent events. In fact, the highest values of $h_s$ are observed in the Ionian coast, while the highest values of $h_{s1}$ observed in the two macro-areas are very similar to each other. All this is also highlighted in the analysis of the characteristic parameters. In fact, the parameter $u$, which shows how the significant wave heights vary as the return period varies and is inversely proportional to their ratio, is generally lower in the Tyrrhenian coast than in the Ionian coast. Instead, the parameter $w$, which is also called the scale factor and is directly proportional to the values of significant wave height, is generally greater in the Ionian coast than in the Tyrrhenian coast. The most intense combination is that of low $u$ and high $w$, as it means that the significant wave height values are high, and they increase significantly as the return period increases. Furthermore, the parameter $a_{10}$, which represents the average value of the maximum significant wave heights of the sample consisting of the more than 400 most intense MTEs (equal to 10 for the number of years of data, as defined above), is very similar between the two macro-areas. On the other hand, the parameter $b_{10}$, which represents the average value of the durations of the more than 400 most intense MTEs, as defined above, is generally greater in the Ionian Sea. So, the sea storms of the Ionian Sea are generally more persistent than the Tyrrhenian ones. This confirms that, in the Ionian coast, frequent events are often more intense than in the Tyrrhenian coast, while in the Tyrrhenian coast, the significant wave heights increase more as the return period increases, so rare events are generally shorter and more intense than in the Ionian coast.

Even from the point of view of the wave energy content, the two coasts are very different. In fact, the highest values of the total energy flux $\Phi_t$ (Figure 7) are observed in the Ionian coast. On the other hand, the values of energy flux of the main sector $\Phi_{MS}$ (Figure 8) observed in the Tyrrhenian coast are generally much greater than those observed in the Ionian coast. Furthermore, the main sectors of the Tyrrhenian coast are all associated with directions coming from the west and the north-west, with small variations between one point and another. On the other hand, in the Ionian coast, the main sectors are associated with directions coming from both the north and the south-east, with considerable variations between one point and another. Additionally, the secondary and tertiary sectors often have energy flux values $\Phi_{SS}$ and $\Phi_{TS}$ slightly lower than $\Phi_{MS}$. In addition to this, it is highlighted

that in no Tyrrhenian points are there secondary and tertiary sectors (Figure 9). Therefore, even from a wave energy content point of view, the Tyrrhenian coast is more homogeneous than the Ionian coast. In fact, in the Tyrrhenian Sea, the wave motion generally comes from a narrow range of directions. Along these directions, the wave energy content is high and much greater than the Ionian Sea, but the total energy content is often lower than the Ionian Sea. Instead, the Ionian Sea is characterized by wave motion coming from a wide range of directions; however, along these directions, the wave energy content is lower than that of the Tyrrhenian. All this is also highlighted by the relationship between the main sector energy flux value and the total energy flux value $\Phi_{MS}/\Phi_t$, which shows the percentage of total energy flux coming from the main sector only. In fact, in the Tyrrhenian Sea, this ratio is always higher than 20%, while in the Ionian Sea, this ratio is always lower than 15%, and the points where the lower values are observed are those characterized by the greatest total energy flux. Therefore, in these points, the total energy content is high but disperses along various directions. The points located in the northern part of the Gulf of Taranto are an exception, where the ratio $\Phi_{MS}/\Phi_t$ is higher than 20% as in the Tyrrhenian Sea. In fact, due to the morphology of the coast, in these points, the intense wave motion can only come from the south-east directions, from the mouth of the Gulf, as the other directions are characterized by modest fetch extensions.

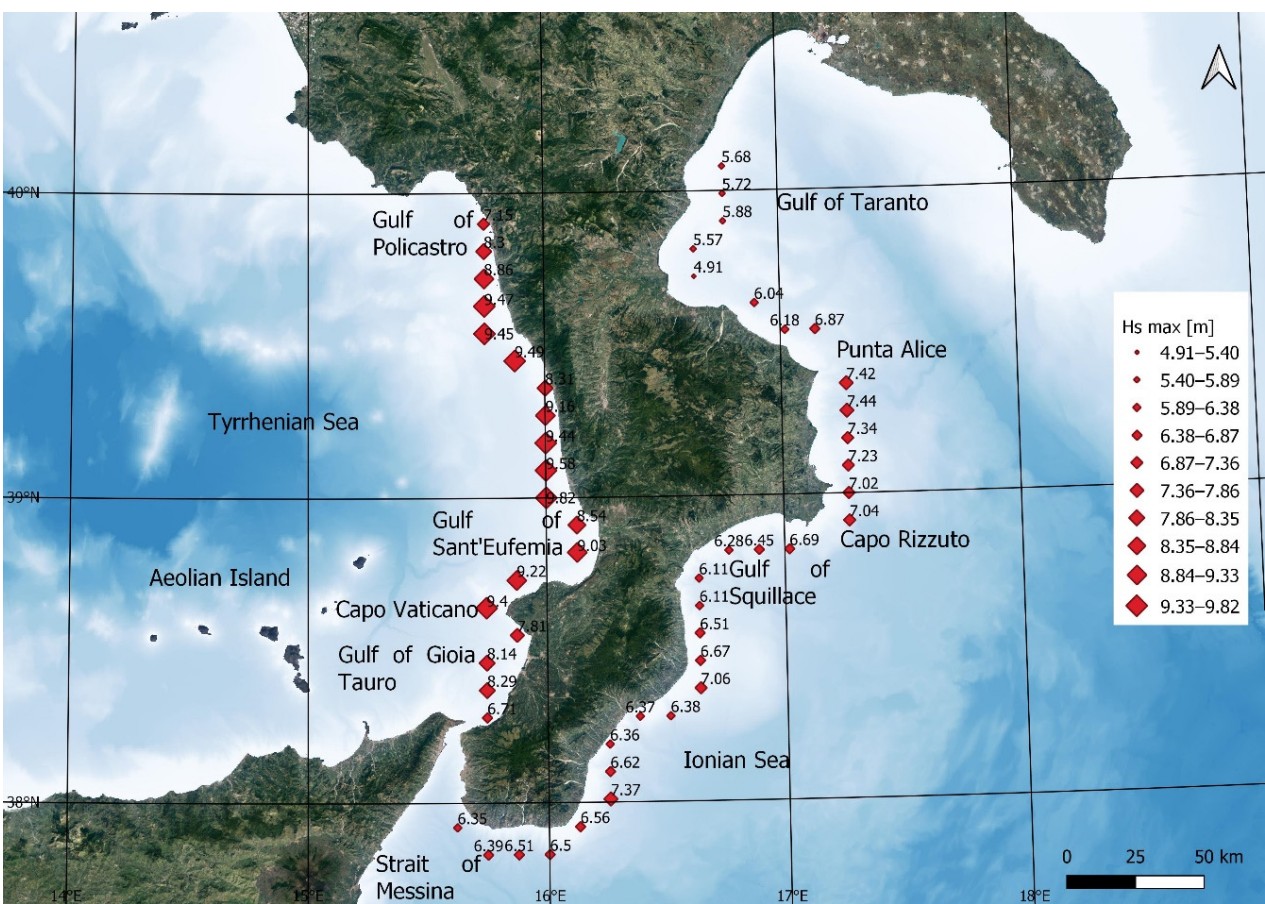

**Figure 3.** Maximum significant wave height $h_{s,max}$ values. The range between the minimum and the maximum values has been divided into 10 intervals of the same width.

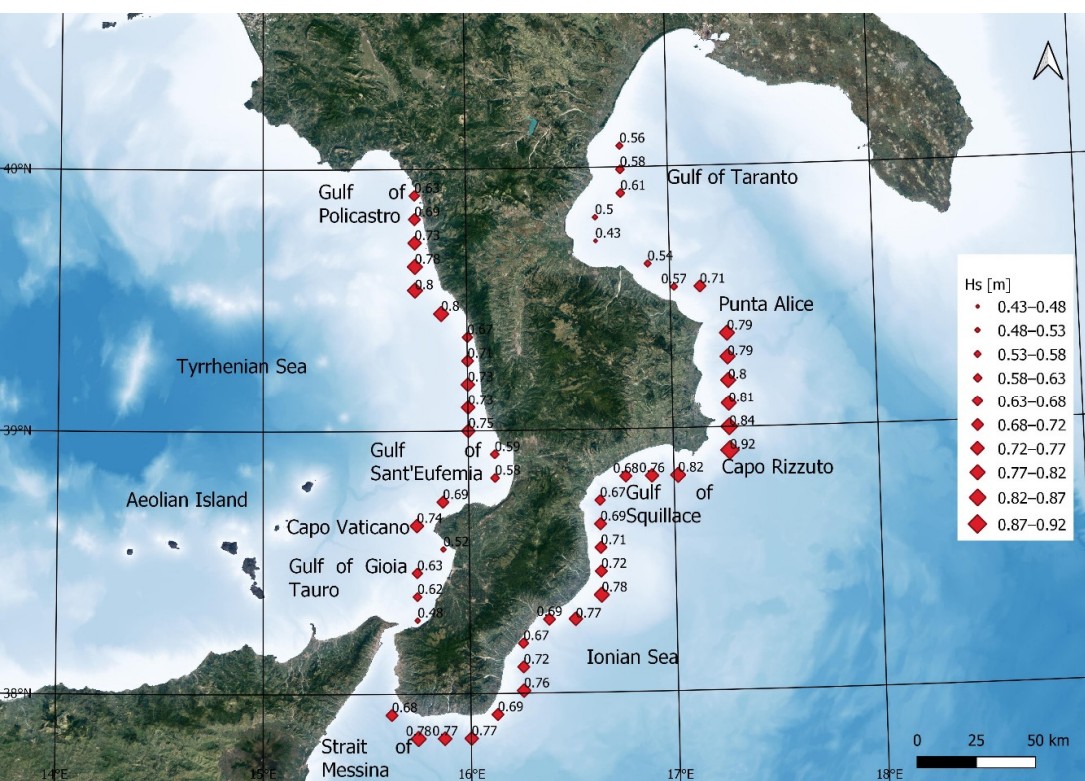

**Figure 4.** Average significant wave height $h_s$ values. The range between the minimum and the maximum values has been divided into 10 intervals of the same width.

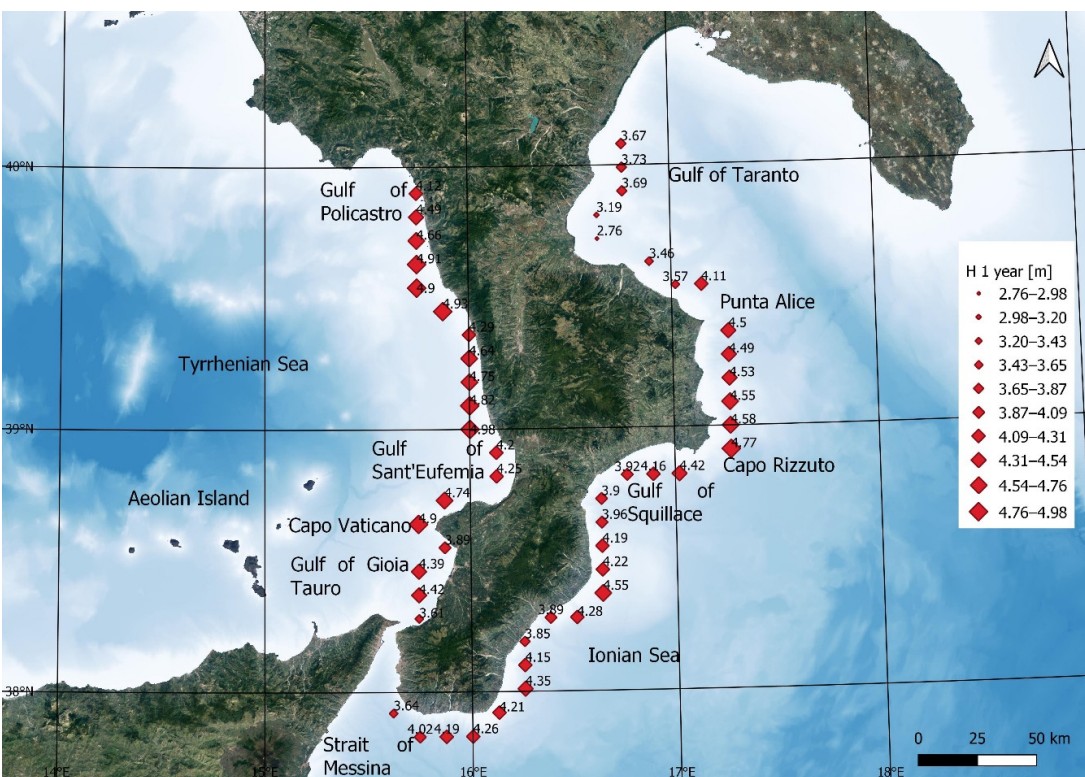

**Figure 5.** Significant wave height of return period of 1 year $h_{s1}$ values. The range between the minimum and the maximum values has been divided into 10 intervals of the same width.

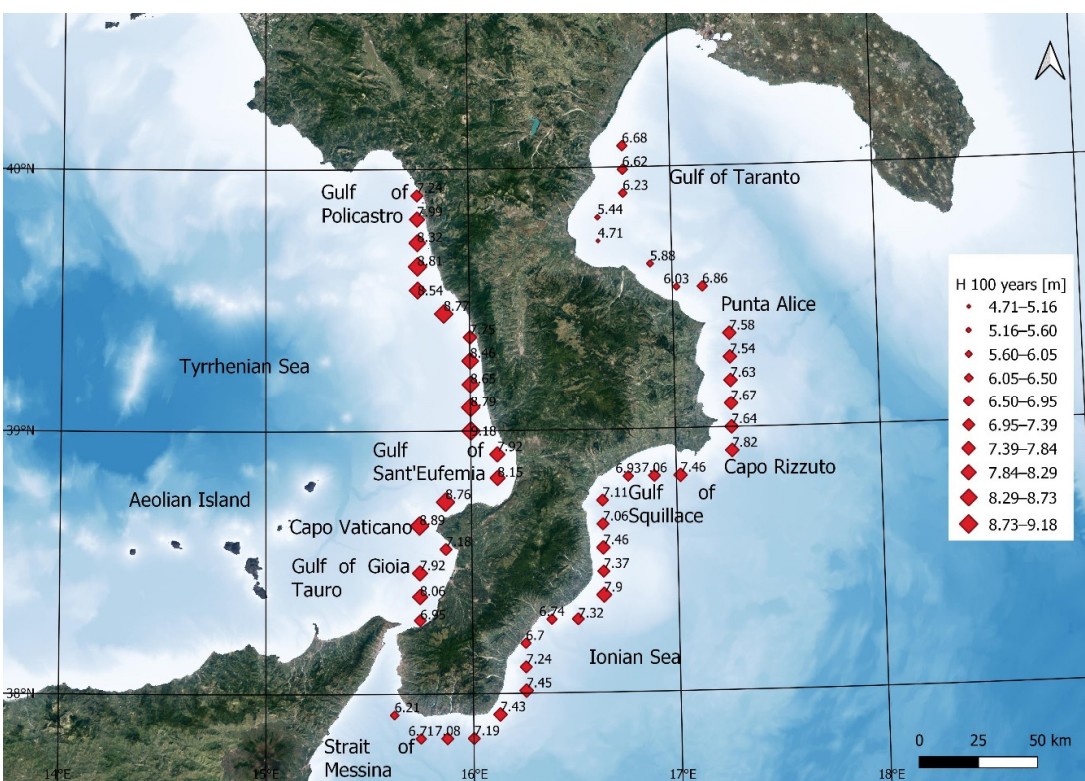

**Figure 6.** Significant wave height of return period of 100 years $h_{s100}$ values. The range between the minimum and the maximum values has been divided into 10 intervals of the same width.

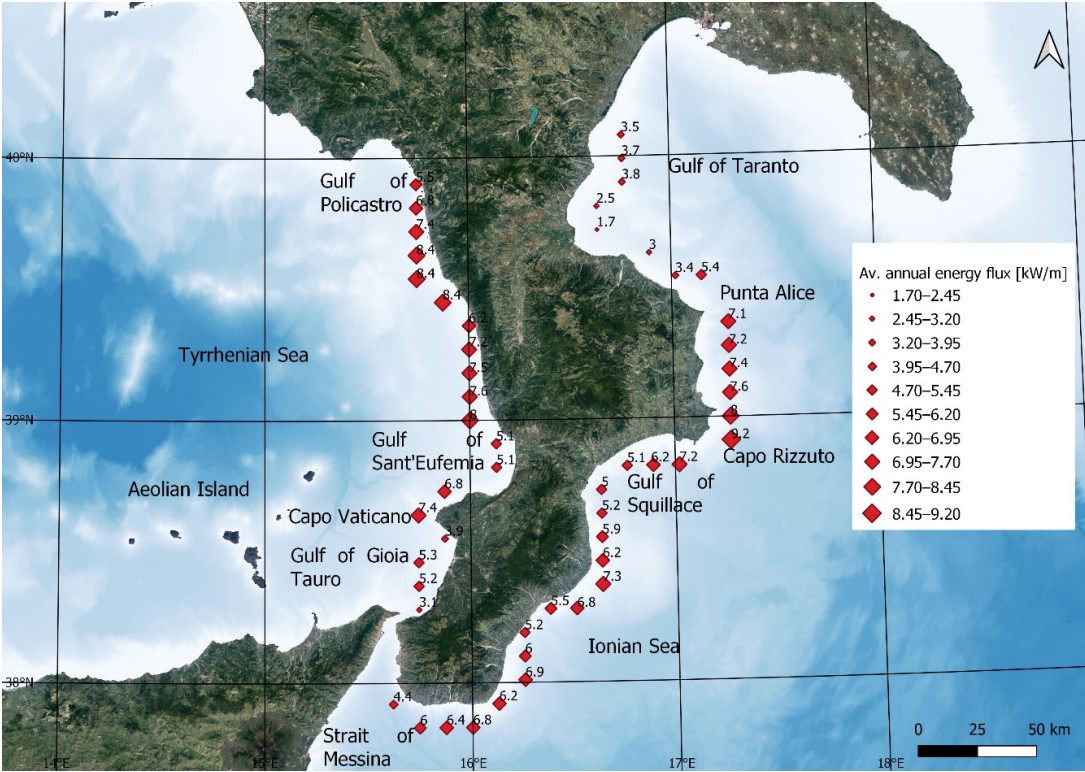

**Figure 7.** Average annual energy flux $\Phi_t$ values. The range between the minimum and the maximum values has been divided into 10 intervals of the same width.

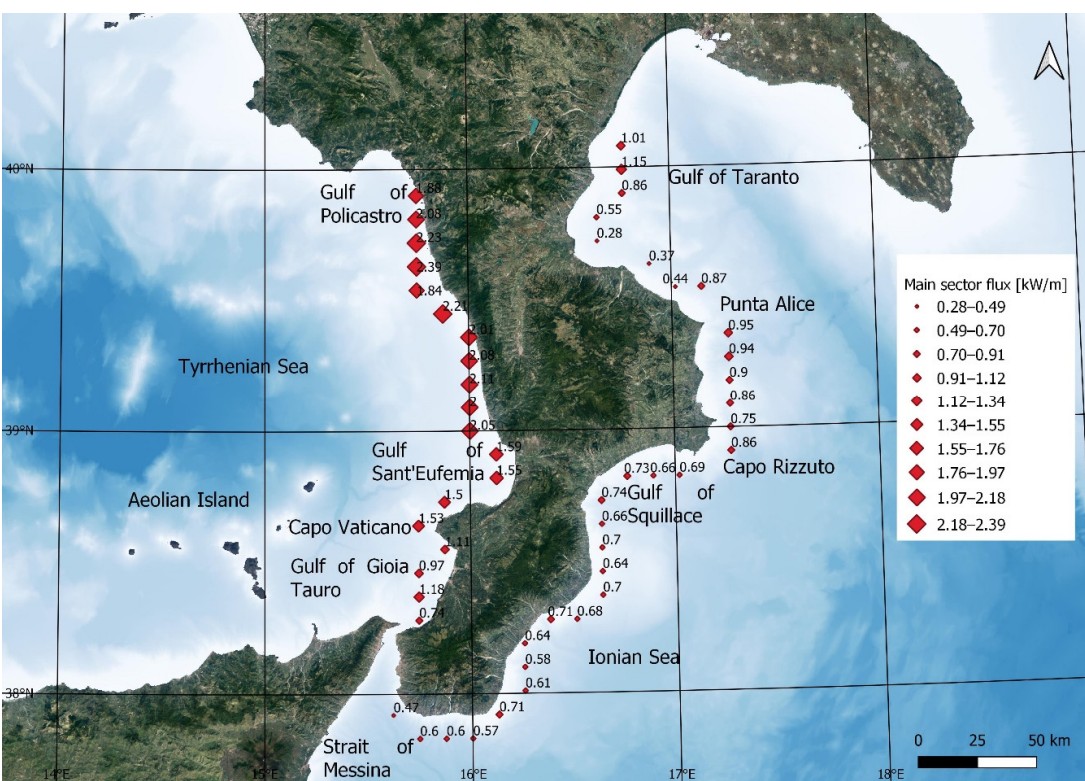

**Figure 8.** Average annual energy flux of main sectors $\Phi_{MS}$ values. The range between the minimum and the maximum values has been divided into 10 intervals of the same width.

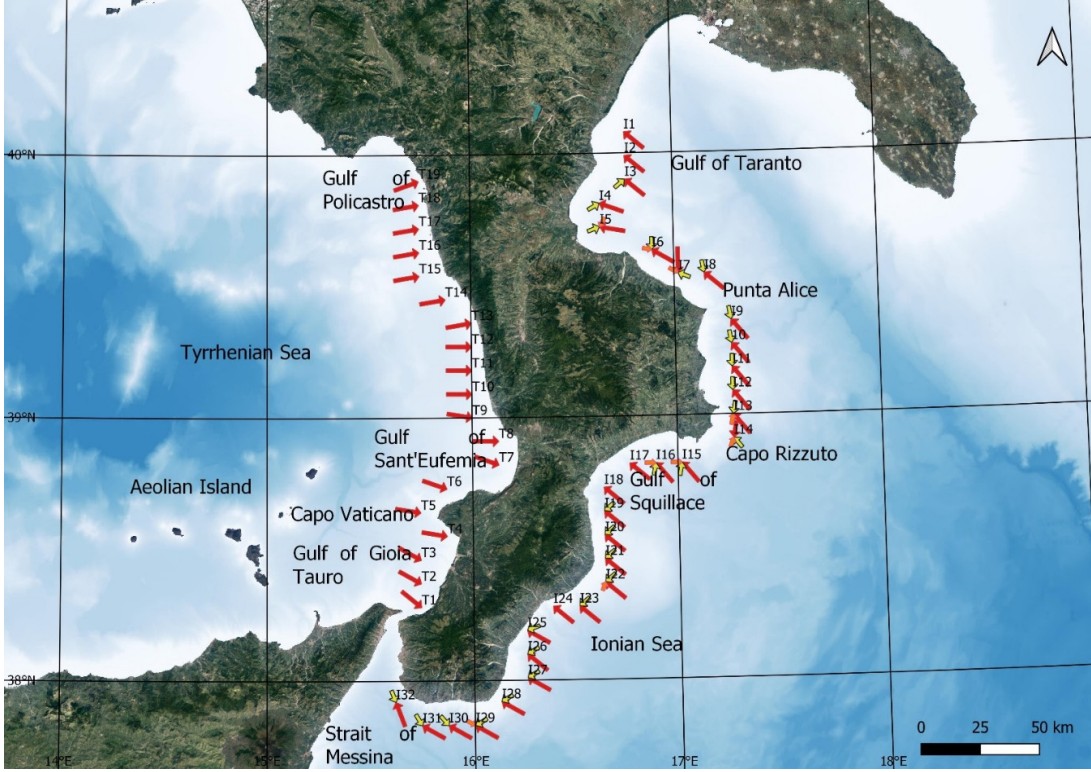

**Figure 9.** Main, secondary, and tertiary energy sector directions. Legend: main sectors *MS* are in red, secondary sectors *SS* are in yellow, and tertiary sectors *TS* are in orange.

The seasonal analysis showed that, in autumn in most of the Calabrian coasts, except for the Gulf of Taranto and the southern Ionian coast, the maximum values of significant wave height $h_{s,max}$ are higher than in winter. This phenomenon is related to the high sea water temperature in the autumn months that favors the formation of TLC. Another important result of the seasonal analysis is that, in almost the entire Ionian coast, the $h_{s,max}$ values in the summer are similar to or greater than in the spring, while in the Tyrrhenian, this is not the case. Therefore, the seasonal variations of $h_{s,max}$ are much greater in the Tyrrhenian coast than in the Ionian coast. Consequently, the danger of the Tyrrhenian Sea about wave climate is generally limited to autumn and winter, while in the Ionian Sea, it is extended to the whole year. Furthermore, in many parts of the Ionian Sea, the main summer sectors *MS* are very different from those of the other seasons and are often associated with directions coming from the west and the north-west. However, it is a wave motion not directed from sea to land but, vice versa, directed from land to sea and generated by winds acting on the Tyrrhenian Sea that cross the mountain reaching the Ionian coasts. This phenomenon is most evident in the plains of Sibari, Lamezia and Gioia Tauro and at the southern mouth of the Strait of Messina. Furthermore, the particularities of two coastal areas should be highlighted. The first area is between the promontories of Punta Alice and Capo Rizzuto, and the second area is in the innermost part of the Gulf of Taranto, near point I5. In the first area, many parameters reach the highest values of the whole Ionian coast and, in the case of the total energy flux $\Phi_t$, reach the highest values of the whole of Calabria. All this can be mainly related to the geographical position of this area, which is the easternmost point of Calabria and is exposed to fetches of the order of hundreds of kilometers along most directions, including those coming from the Gulf of Taranto. On the other hand, in the second area, almost all the analyzed parameters reach the lowest values of the whole of Calabria. Additionally, in this case, all this can be mainly related to the geographical position of this area, for which the fetches have modest lengths along most directions.

The results obtained are fairly in line with those obtained by analyzing the time series recorded by the two buoys of the National Wave Network (RON) present in Calabria. One buoy is in Crotone, on the Ionian Sea near point I13, and another buoy is in Cetraro, in the Tyrrhenian Sea near point T14. Both buoys are characterized by shorter time series than those analyzed, equal to 25 years in Crotone and 15 years in Cetraro, with the latest recording available in 2014 in both buoys. In particular, the maximum significant heights are about 0.5 m lower, and the average heights are about 0.1 m lower than those calculated in the present analysis.

Finally, a comparison was made between this analysis and the main analysis carried out along the Calabrian coast by Barbaro [82], Ferraro et al. [83] and Caloiero et al. [84]. In that of Barbaro, the wave climate was analyzed in 24 points, less than half of the present analysis, with a less extended time series, 20 years, and considering only four parameters. In the study by Ferraro et al., the wave climate was analyzed in several points like that of the present analysis but considering a time series of just over 20 years and estimating only the annual and seasonal energy potential. The study by Caloiero et al. followed that of Ferraro et al., also analyzing significant wave height and energy period and extending the time series to about 40 years. Therefore, the analysis described in this paper is characterized by a spatial and temporal level of detail and by a greater number of parameters analyzed than in the main previous analysis carried out in Calabria.

## 5. Conclusions

This paper described an analysis of the wave climate carried out along the Calabrian coasts. Calabria is a region of the Southern Italy that represents an interesting case study due to its high coastal length, over 700 km, and due to its geomorphological and climatic complexity. Calabria is enclosed by two seas, Ionian and Tyrrhenian, and is characterized by various gulfs and promontories that cause considerable variability in terms of fetch length and wave climate. The analysis described in the paper was carried out in over

50 sample areas, each of them covering an average of 15 km of coastline, by examining over 40 years of wave data and about 20 wave parameters representative of annual and seasonal average and maximum wave conditions. This analysis mainly highlighted that the two Ionian and Tyrrhenian coasts are very different from the wave climate point of view. Indeed, the average and frequent wave conditions are slightly higher in the Ionian coast, while the exceptional wave conditions are much greater in the Tyrrhenian coast. Furthermore, in the Tyrrhenian Sea, the intense wave conditions are concentrated along a few directions, coming mainly from the north-west, and there are no secondary and tertiary sectors. Instead, in the Ionian Sea, the intense wave conditions can come from different directions, varying between north-east and south-east, and in different locations, there are secondary and tertiary sectors. Another difference between the two seas concerns the considerable spatial variability of the wave climate along the Ionian Sea, especially within the Gulf of Taranto and between the Punta Alice and Capo Rizzuto promontories, while the wave climate in the Tyrrhenian Sea is substantially homogeneous spatially. The seasonal analysis showed that, in autumn in most of the Calabrian coasts, except for the Gulf of Taranto and the southern Ionian coast, the maximum values of significant wave height are higher than in winter. Another important result of the seasonal analysis is that the danger of the Tyrrhenian Sea about wave climate is generally limited to autumn and winter, while in the Ionian Sea, it is extended to the whole year.

Finally, the analysis described in this paper is characterized by a spatial and temporal level of detail and by a greater number of parameters analyzed than in the main previous analysis carried out in Calabria. So, it is important to analyze the wave climate with high spatial and parametric detail, especially for complex and highly variable geomorphological and climatic conditions within closed basins and in heavily man-made territories with numerous inhabited centers, infrastructures and archaeological sites exposed to wave motion as in this case study. Furthermore, the paper is of particular interest in the fields of planning and management of the coastal areas and of design of coastal defense works.

**Author Contributions:** Conceptualization, G.F. and G.C.B.; methodology, G.F., G.B. (Giuseppe Barbaro), G.B. (Giovanni Besio), G.C.B., P.M. and P.P.; software, G.F. and G.C.B.; validation, G.F., G.B. (Giuseppe Barbaro), G.B. (Giovanni Besio), G.C.B., P.M. and P.P.; formal analysis, G.F. and G.C.B.; investigation, G.F. and G.C.B.; resources, G.F. and G.C.B.; data curation, G.F. and G.C.B.; writing—original draft preparation, G.F.; writing—review and editing, G.F., G.B. (Giuseppe Barbaro), G.B. (Giovanni Besio), G.C.B., P.M. and P.P.; visualization, G.F.; supervision, G.B. (Giuseppe Barbaro); project administration, G.B. (Giuseppe Barbaro); funding acquisition, G.B. (Giuseppe Barbaro) and P.M. All authors have read and agreed to the published version of the manuscript.

**Funding:** This research was funded by the Public Works Department of Calabria Region.

**Institutional Review Board Statement:** Not applicable.

**Informed Consent Statement:** Not applicable.

**Data Availability Statement:** Not applicable.

**Conflicts of Interest:** The authors declare no conflict of interest. The funders had no role in the design of the study; in the collection, analyses, or interpretation of data; in the writing of the manuscript, or in the decision to publish the results.

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
