# Peer review of "Wave Climate along Calabrian Coasts"

_climate, doi:10.3390/cli10060080_

Round 1

Reviewer 1 Report

The article is headed «Wave climate along Calabrian coasts». Based on the abstract, the peer-reviewed article corresponds to the profile of the Climate journal.

The paper analyzes the wave climate of the southern Italy coast. 50 points were selected along the coast, for which wave parameters were calculated.

Comments.

  1. Line 31. It states «the wave climate is generally proportional to both the wind speed and the length of the fetch». But that is not all. The wind duration, the airflow stability, as well as the depth, i.e. the bathymetric features of the region are of great importance. And here we encounter an extremely interesting point. Fig. 1 shows the object of research, Fig. 2 shows the points position. But there is no bathymetry! Why? From the introduction, reader learns many interesting things about the mountains’ height, about precipitation, but not a single word about bathymetry features. Aren't there any? Let us examine a map, for example, Navionics:

It can be seen the Tyrrhenian Sea is shallower than the Ionian Sea. In addition, the west Calabria coast is in the wave shadow of an island chain. The shelf width is different. Doesn’t it matter?

  1. Line 48. «Wave climate depends on many parameters, the main ones being significant wave height…». Wave climate does not depend on parameters, it is determined by those parameters.
  2. It is said that Metocean database is used for analysis. The sample was taken over the last 40 years. It is necessary to specify: (1) start and end years, (2) time step (one hour, three hours) - this is very important. In addition, the authors point to the geographic features of the wind conditions, namely the local Scirocco and Mistral winds. Does the used atmospheric model reproduce these local features sufficiently? What wave model is used?
  3. It is stated that the selected points are about 15 km from the coast. At the same time on Fig. 2 we can see clearly that they are located in the nodes of a regular grid. A cursory analysis allows for a statement that some points are also located 5 km from the shore (maybe even closer). Thus, differences in the wave transformation patterns may arise. Considering the shallow water, breaking waves are possible at some points, it obviously affects the statistics. How is the wave data compared for them?
  4. The numbering of points is not entirely the best choice. Wouldn't it be more convenient, for example, to designate the points of the Ionian Sea as I1, I2… and the Tyrrhenian Sea as T1, T2...?
  5. Why is the Weibull representation (formula 1) used? There are recognized POT, AMS methods. If Weibull is used, it must be shown that the original data obeys the Weibull's law. Why do we need a repeatability of every 500 years? Usually 100 years is enough (physically).
  6. It is said that «The energy flux of each sector is the sum of the energy flux of each sea state from that sector…». If it is a sum, why the dimension is N/s? In any case, it would be more convenient to represent in kWt. At the same time, the flux values from Table 2 seem to be too small for the "sums". Most likely they are average values (of course, this begs for the question, what is the average - the annual average?).
  7. Figures. Why is the data in Figures 3-8 presented in gradations? After all, each point corresponds to a certain number.

Figures 9-11. It is not quite clear what are the secondary and tertiary sectors needed for. If the authors find them informative, wouldn’t it be more appropriate to display them not with numbers, but with arrows in one figure, for example, like this:

In summary.

A lot of work has been done and it will certainly be of an interest to subject matter specialists. At the same time, I would like to highlight some points, to which the authors should pay attention in order to prepare a more informative article:

  1. How well does the database reflect the true situation? It is said that there are two wave buoys in the area (or they were there before, it does not matter). Why not to compare the data from buoy with the database?
  2. It is also said that wind waves and swell are separated in the database. That is great! It would be interesting to get separate statistics.
  3. The data from Table 2 and in Fig. 3-4 (ratio of maximum and average values of Hs, as well as small periods) show that we are most likely dealing with a pronounced seasonal distribution of storm activity. In other words, why not to calculate the statistics separately by seasons or by months.

Author Response

Comment

The article is headed «Wave climate along Calabrian coasts». Based on the abstract, the peer-reviewed article corresponds to the profile of the Climate journal.

The paper analyzes the wave climate of the southern Italy coast. 50 points were selected along the coast, for which wave parameters were calculated.

Reply

Dear Reviewer, many thanks for your helpful comments.

Comment

Line 31. It states «the wave climate is generally proportional to both the wind speed and the length of the fetch». But that is not all. The wind duration, the airflow stability, as well as the depth, i.e. the bathymetric features of the region are of great importance. And here we encounter an extremely interesting point. Fig. 1 shows the object of research, Fig. 2 shows the points position. But there is no bathymetry! Why? From the introduction, reader learns many interesting things about the mountains’ height, about precipitation, but not a single word about bathymetry features. Aren't there any? Let us examine a map, for example, Navionics:

It can be seen the Tyrrhenian Sea is shallower than the Ionian Sea. In addition, the west Calabria coast is in the wave shadow of an island chain. The shelf width is different. Doesn’t it matter?

Reply

The wind duration, the airflow stability and the bathymetry were mentioned in the Introduction. In paragraph 2.1 a detailed analysis of the bathymetry around the Calabrian coasts has been added, starting from of the bathymetry available on the European Marine Observation and Data Network (EMODnet) portal (https://www.emodnet-bathymetry.eu/).

Comment

Line 48. «Wave climate depends on many parameters, the main ones being significant wave height…». Wave climate does not depend on parameters, it is determined by those parameters.

Reply

The words “depends on” have been replaced by “is determined by”.

Comment

It is said that Metocean database is used for analysis. The sample was taken over the last 40 years. It is necessary to specify: (1) start and end years, (2) time step (one hour, three hours) - this is very important. In addition, the authors point to the geographic features of the wind conditions, namely the local Scirocco and Mistral winds. Does the used atmospheric model reproduce these local features sufficiently? What wave model is used?

Reply

Information on start and end years, on time step, on atmospheric model and on wave model has been added.

Comment

It is stated that the selected points are about 15 km from the coast. At the same time on Fig. 2 we can see clearly that they are located in the nodes of a regular grid. A cursory analysis allows for a statement that some points are also located 5 km from the shore (maybe even closer). Thus, differences in the wave transformation patterns may arise. Considering the shallow water, breaking waves are possible at some points, it obviously affects the statistics. How is the wave data compared for them?

Reply

The 15 km mentioned do not refer to the distance from the coast but to the distance along the coast between one point and another. Furthermore, all points are located on deep water and this information has been added.

Comment

The numbering of points is not entirely the best choice. Wouldn't it be more convenient, for example, to designate the points of the Ionian Sea as I1, I2… and the Tyrrhenian Sea as T1, T2...?

Reply

The previous numbering reproduced the original MeteOcean number, however it has been modified according to your indications.

Comment

Why is the Weibull representation (formula 1) used? There are recognized POT, AMS methods. If Weibull is used, it must be shown that the original data obeys the Weibull's law. Why do we need a repeatability of every 500 years? Usually 100 years is enough (physically).

Reply

Among the various distributions of the probability of exceeding, Weibull was chosen as appropriate for the Mediterranean Sea. Furthermore, some examples have been cited where this distribution has been applied in the Italian seas. Significant wave height of return period of 500 years has been replaced by significant wave height of return period of 100 years.

Comment

It is said that «The energy flux of each sector is the sum of the energy flux of each sea state from that sector…». If it is a sum, why the dimension is N/s? In any case, it would be more convenient to represent in kWt. At the same time, the flux values from Table 2 seem to be too small for the "sums". Most likely they are average values (of course, this begs for the question, what is the average - the annual average?).

Reply

Previously it was considered N/s as a unit of measurement to emphasize the differences in values between the sample areas. As required, the obtained values were converted into kW/m. Furthermore, it has been clarified that these are annual average values.

Comment

Figures. Why is the data in Figures 3-8 presented in gradations? After all, each point corresponds to a certain number.

Reply

The data of these figures have been presented in gradation to emphasize the differences between the different Calabrian coastal areas.

Comment

Figures 9-11. It is not quite clear what are the secondary and tertiary sectors needed for. If the authors find them informative, wouldn’t it be more appropriate to display them not with numbers, but with arrows in one figure, for example, like this:

Reply

The three figures have been unified and the various sectors have been indicated with arrows, as suggested.

Comment

In summary.

A lot of work has been done and it will certainly be of an interest to subject matter specialists. At the same time, I would like to highlight some points, to which the authors should pay attention in order to prepare a more informative article:

How well does the database reflect the true situation? It is said that there are two wave buoys in the area (or they were there before, it does not matter). Why not to compare the data from buoy with the database?

Reply

In the Discussions section, a comparison between the wave climate parameters obtained by analyzing the data of the two buoys and the wave climate parameters obtained by analyzing the MeteOcean database has been added.

Comment

It is also said that wind waves and swell are separated in the database. That is great! It would be interesting to get separate statistics.

Reply

Among the parameters available in the MeteOcean database are significant height, period and direction aggregated or separated between wind wave and first and second swell. However, for the purposes of this analysis only the aggregated data were acquired. However, a seasonal analysis has been added.

Comment

The data from Table 2 and in Fig. 3-4 (ratio of maximum and average values of Hs, as well as small periods) show that we are most likely dealing with a pronounced seasonal distribution of storm activity. In other words, why not to calculate the statistics separately by seasons or by months.

Reply

A seasonal analysis was performed where the following parameters were calculated (where the i-th season has been indicated with subscripts W for win-ter, SP for spring, SU for summer and A for autumn): seasonal maximum significant wave height hs,max,i; seasonal average significant wave height hs,i; seasonal average annual energy flux Фt,i; seasonal main sector MSi.

Reviewer 2 Report

This manuscript is devoted to examining the wave conditions using the wave database by the MeteOcean Group. The explanation is insufficient, monotonous, and boring, and therefore the following revisions are required to improve the manuscript:

  • The authors just use the existing wave database, so the originality is unclear. The authors should emphasize the originality more clearly together with the clear explanation on the relationship with the previous research focusing on the same area.
  • The authors mention coastal erosion process, coastal and river sedimentary balance, etc. However, the relationship with the wave conditions is unclear because the authors do not present the change in the shoreline position, beach bathymetry, etc. The authors should explain why they mention coastal erosion process etc.
  • All figures are similar using satellite imagery. The authors should improve the figures to make the relationship with the discussion clearer.

Author Response

Comment

This manuscript is devoted to examining the wave conditions using the wave database by the MeteOcean Group. The explanation is insufficient, monotonous, and boring, and therefore the following revisions are required to improve the manuscript:

The authors just use the existing wave database, so the originality is unclear. The authors should emphasize the originality more clearly together with the clear explanation on the relationship with the previous research focusing on the same area.

The authors mention coastal erosion process, coastal and river sedimentary balance, etc. However, the relationship with the wave conditions is unclear because the authors do not present the change in the shoreline position, beach bathymetry, etc. The authors should explain why they mention coastal erosion process etc.

All figures are similar using satellite imagery. The authors should improve the figures to make the relationship with the discussion clearer.

Reply

Dear Reviewer, many thanks for your helpful comments.

The manuscript has been rewritten in many parts, clarifying the originality with respect to previous studies, especially those of Barbaro (2016), Ferraro et al. (2017) and Caloiero et al. (2019).

Discussions have been improved by adding various topics such as coastal erosion, anthropogenic pressure, river sediment transport, sea storms, coastal defense works etc.

The figures have been improved by adding bathymetry and labels with the values of the various parameters to improve readability.

Reviewer 3 Report

The manuscript provides a good analysis of wave characteristics in southern Italy and is expected to be a valuable data set. In particular, the paper is unique in that it mentions, with many quantitative data, regional differences in the wave's frequent and rare events. The results are described in detail, but I consider that a more in-depth Discussion and corresponding modifications to the Abstract and Introduction are needed.

Abstract: I found the structure of the Abstract problematic. While there is a lot of text describing the background and the study site, there is a short description of the methods, results, and discussion. It needs to be rewritten significantly.

Lines 12–13: You mention anthropogenic pressure and sediment budgets, but I don't think there was any mention of these in the Results or Discussion; it would be desirable to revise the Results or Discussion or remove this section from the Abstract.

Line 16: I did not understand well the meaning of high coastal length. Can you use another word to describe it?

Line 31: Isn't fetch length a more common expression than fetches length?

Lines 60–67: The objectives of this study should be more clearly stated. After stating the purpose, the details of what was conducted should be described.

Line 77, 81: Place names such as Libyan Sea and Island of Elba are mentioned, but readers unfamiliar with Italian place names may be confused about the exact location. The place names need to be appropriately marked on the map.

Line 78: Bathymetry is mentioned. Is it possible to add bathymetry contours to Fig. 1? Or, it could be indicated by changing the color of the ocean depending on the depth of the water.

Lines 96–105: Are any of these geology topics mentioned in the Discussion? I think most of the text could be removed.

Lines 106–123: The information may be important to understand the characteristics of the study area, but I felt that it was too voluminous. If there is information that is not so relevant to Results or Discussion, you might consider removing it.

Line 135: It says "third-generation model", but what specific model calculation results did you refer to? (e.g., WAVEWATCH III, WAM, SWAN)

Discussion: It is desirable to correlate the results of wave climate analysis with local coastal management. In the current manuscript, only the wave climate results are presented.

Discussion: Is there any reason why there are no citations to previous studies in this section? If the emphasis is on peculiarity, it is necessary to specify how wave climate differs from other countries and regions.

Author Response

Comment

The manuscript provides a good analysis of wave characteristics in southern Italy and is expected to be a valuable data set. In particular, the paper is unique in that it mentions, with many quantitative data, regional differences in the wave's frequent and rare events. The results are described in detail, but I consider that a more in-depth Discussion and corresponding modifications to the Abstract and Introduction are needed.

Reply

Dear Reviewer, many thanks for your helpful comments.

Comment

Abstract: I found the structure of the Abstract problematic. While there is a lot of text describing the background and the study site, there is a short description of the methods, results, and discussion. It needs to be rewritten significantly.

Reply

The abstract has been restructured according to your comments.

Comment

Lines 12–13: You mention anthropogenic pressure and sediment budgets, but I don't think there was any mention of these in the Results or Discussion; it would be desirable to revise the Results or Discussion or remove this section from the Abstract.

Reply

These topics have been added to the Discussion.

Comment

Line 16: I did not understand well the meaning of high coastal length. Can you use another word to describe it?

Reply

The word “high” has been replaced by “notable”.

Comment

Line 31: Isn't fetch length a more common expression than fetches length?

Reply

This correction was made.

Comment

Lines 60–67: The objectives of this study should be more clearly stated. After stating the purpose, the details of what was conducted should be described.

Reply

This section has been rewritten and the objectives of the study have been better clarified.

Comment

Line 77, 81: Place names such as Libyan Sea and Island of Elba are mentioned, but readers unfamiliar with Italian place names may be confused about the exact location. The place names need to be appropriately marked on the map.

Reply

In the small panel of Fig. 1 the labels Mediterranean Sea, Libyan Sea, Corsica, and Sardinia have been added. The Island of Elba label has not been added as it is too small, so the sentence where it was mentioned has been changed.

Comment

Line 78: Bathymetry is mentioned. Is it possible to add bathymetry contours to Fig. 1? Or, it could be indicated by changing the color of the ocean depending on the depth of the water.

Reply

The bathymetry around the Calabrian coasts has been added in paragraph 2.1, starting from of the bathymetry available on the European Marine Observation and Data Network (EMODnet) portal (https://www.emodnet-bathymetry.eu/). Also, Fig. 1 has been modified by adding the bathymetry with a colored graduated scale and highlighting the bathymetry -100, -500 and 1000 m.

Comment

Lines 96–105: Are any of these geology topics mentioned in the Discussion? I think most of the text could be removed.

Reply

This topic has been added to the Discussion.

Comment

Lines 106–123: The information may be important to understand the characteristics of the study area, but I felt that it was too voluminous. If there is information that is not so relevant to Results or Discussion, you might consider removing it.

Reply

These topics have been added to the Discussion.

Comment

Line 135: It says "third-generation model", but what specific model calculation results did you refer to? (e.g., WAVEWATCH III, WAM, SWAN)

Reply

A description of this model has been added.

Comment

Discussion: It is desirable to correlate the results of wave climate analysis with local coastal management. In the current manuscript, only the wave climate results are presented.

Reply

This topic has been added to the Discussion.

Comment

Discussion: Is there any reason why there are no citations to previous studies in this section? If the emphasis is on peculiarity, it is necessary to specify how wave climate differs from other countries and regions.

Reply

About 15 citations were added in the discussions.

Round 2

Reviewer 1 Report

Thanks. The authors did a good job of improving the article.

Reviewer 2 Report

The authors revised the manuscript according to the reviewers’ comments.

Reviewer 3 Report

In response to my peer review comments, I confirm that the figures and text have been appropriately revised. I am sure that many revisions have made your paper of higher quality.